# Structural Study of Model Rhodium(I) Carbonylation Catalysts Activated by Indole-2-/Indoline-2-Carboxylate Bidentate Ligands and Kinetics of Iodomethane Oxidative Addition

**Mohammed A. E. Elmakki** [1,2,*], **Orbett Teboho Alexander** [2], **Gertruida J. S. Venter** [2], **Johan Andries Venter** [2,*] **and Andreas Roodt** [2,*]

1    Department of Chemistry, Omdurman Islamic University, P.O. Box 382, Omdurman 14415, Sudan
2    Department of Chemistry, University of the Free State, P.O. Box 339, Bloemfontein 9300, South Africa
*    Correspondence: azeez77777@gmail.com (M.A.E.E.); venterja@ufs.ac.za (J.A.V.); aroodta@gmail.com (A.R.)

**Abstract:** The rigid-backbone bidentate ligands Indoline-2-carboxylic acid (IndoliH) and Indole-2-carboxylic acid (IndolH) were evaluated for rhodium(I). IndoliH formed [Rh(Indoli)(CO)(PPh$_3$)] (**A2**), while IndolH yielded the novel dinuclear [Rh$^1$(Indol')(CO)(PPh$_3$)Rh$^2$(CO)(PPh$_3$)$_2$] (**B2**) complex (Indol' = Indol$^{2-}$), which were characterized by SCXRD. In **B2**, the Rh$^1$(I) fragment [Rh$^1$(Indol')(CO)(PPh$_3$)] (bidentate *N,O*-Indol) exhibits a square-planar geometry, while Rh$^2$(I) shows a 'Vaska'-type trans-[O-Rh$^2$(PPh$_3$)$_2$(CO)] configuration (bridging the carboxylate 'oxo' O atom of Indol$^{2-}$). The oxidative addition of MeI to **A2** and **B2** via time-resolved FT-IR, NMR, and UV/Vis analyses indicated only Rh(III)-*alkyl* species (**A3**/**B3**) as products (no migratory insertion). Variable temperature kinetics confirmed an associative mechanism for **A2** via an equilibrium-based pathway ($\Delta H^{\neq} = (21 \pm 1)$ kJ mol$^{-1}$; $\Delta S^{\neq} = (-209 \pm 4)$ J K$^{-1}$mol$^{-1}$), with a smaller contribution from a reverse reductive elimination/solvent pathway. The dinuclear complex **B2** showed the oxidative addition of MeI *only* at Rh$^1$(I), which formed a Rh(III)-alkyl, but cleaved the bridged Rh$^2$(I) site, yielding trans-[Rh$^1$(PPh$_3$)$_2$(I)(CO)] (**5B**) as a secondary product. A significantly smaller negative activation entropy [$\Delta H^{\neq} = (73.0 \pm 1.2)$ kJ mol$^{-1}$; $\Delta S^{\neq} = (-21 \pm 4)$ J K$^{-1}$mol$^{-1}$] via a more complex/potential interchange mechanism (the contribution of $\Delta S^{\neq}$ to the Gibbs free energy of activation, $\Delta G^{\neq}$, only $\pm 10\%$) was inferred, contrary to the entropy-driven oxidative addition of MeI to **A2** (the contribution of $\Delta S^{\neq}$ to $\Delta G^{\neq} \pm 75\%$).

**Keywords:** kinetics; crystal structures; iodomethane; oxidative addition; rhodium; Indole-2-carboxylate; Indoline-2-carboxylate; dinuclear complex

## 1. Introduction

Rhodium complexes represent some of the most widely propagated industrial homogeneous catalysts, and classic examples include olefin hydrogenation by Wilkinson's catalyst [RhCl(PPh$_3$)$_3$], the carbonylation of methanol to produce acetic acid by using the catalyst precursor [Rh(CO)$_2$I$_2$]$^-$ in the Monsanto process, and the hydroformylation of alkenes to produce aldehydes by Union Carbide using [RhHCO(PPh$_3$)$_2$] [1,2]. The catalytic cycles are defined by several fundamental reactions, of which oxidative addition is a prime example, and are prominent in the mechanistic scheme of a number of these applied processes. Thus, the formations of Rh$^{III}$-alkyl and Rh$^{III}$-acyl entities are important reaction products in a number of these prominent catalytic processes and has prompted much research aiming to optimize systems.

The Rh catalyst precursor in the Monsanto process, [RhI$_2$(CO)$_2$]$^-$ contains only 'simple' iodido ancillary ligands [1,2]. Since the catalytic reactivity of rhodium(I) complexes is controlled by the nature of the ligands around the metal center, manipulating the electro-steric properties of these groups allows the reaction rates of two essential steps in homogeneous catalysts, such as oxidative addition and migratory insertion, to be tuned. The catalyst precursor [Rh$^I$(I)$_2$(CO)$_2$]$^-$ in the Monsanto process is relatively unstable and may

thus be manipulated by introducing different bidentate ligands, generating, for example, [Rh$^{I}$(*L,L'*-Bid)(CO)$_2$] complexes (*L,L'*-Bid$^-$, a monocharged bidentate ligand, and L and L', coordinating donor atoms).

In addition, depending on the reaction conditions, monodentate (typically tertiary phosphine entities) and non-symmetric bidentate (*L,L'*-Bid) ancillary non-labile ligands with different donor atoms (e.g., O, N, S, P, Se) may be visualized to engineer pre-selected additional characteristics in new catalyst analogues. Thus, one of the carbonyl ligands in the precursor [Rh$^{I}$(*L,L'*-Bid)(CO)$_2$] may additionally be replaced by a monodentate tertiary phosphine, arsine, or stibine ligand to form [Rh$^{I}$(*L,L'*-Bid)(CO)(AX$_3$)] complexes (where A = P, As, and Sb, and X$_3$ = Ph$_3$, Ph$_2$Cy, PhCy$_2$, Cy$_3$, etc.). Moreover, tertiary phosphines, when employed in conjunction with the bidentate ligands, allows for further synergistic tuning of the electronic and steric accessibility of the metal center to a model catalyst [3,4]. In this regard, platinum group metals [5,6], particularly rhodium(I) complexes, and their oxidative addition (Rh$^{III}$-alkyl) and migratory insertion products (which form Rh$^{III}$-acyl entities) have been extensively studied in the past few decades [4,7–14].

Our group focuses on structure/reactivity relationships in model catalytic systems; therefore, we extensively use X-ray crystallography as an integral research component, which is supplemented by IR and multi-nuclear NMR spectroscopy (including time-resolved) to study applicable systems. Different examples have been presented in the past, which include crystallographic studies on [Rh$^{I}$(*L,L'*-Bid)(CO)(PPh$_3$)] complexes that revealed that the reaction between [Rh$^{I}$(*L,L'*-Bid)(CO)$_2$] with PPh$_3$ yields only one isomer that crystallizes from solution [4,15–17]. The only example wherein both isomers crystallized in the same unit cell has been reported for the [Rh$^{I}$(BA)(CO)(PPh$_3$)] complex (BA = benzoylacetonate) [18].

Thus, we present herein examples of rigid-backbone bidentate ligands based on the Indole framework, such as Indole-2-carboxylic acid and Indoline-2-carboxylic acid, which, due to its electronic and relative robust structure, warrants additional and extended investigation. It produced surprising new compounds and provide us with insight into both a monomeric (**A2**) and a dinuclear (**B2**) reactant and the kinetic behavior during the iodomethane (MeI) oxidative addition, including the associated energies therewith. An added advantage resulting from these ligand systems is that migratory insertion is eliminated, which significantly simplifies the interpretation of the reaction mechanism without the interference of any side isomerization reactions.

## 2. Results

### 2.1. Synthesis of Compounds

Indole-2-carboxylate (IndolH = Indole-2-carboxylic acid) and Indoline-2-carboxylate (IndoliH = Indoline-2-carboxylic acid) are known ligands for transition metals. Both act as bidentate ligands by undergoing a mono-dissociation of the carboxylic acid proton, as shown in Scheme 1. However, Indole-2-carboxylic acid showed a two-proton dissociation, as illustrated in Scheme 1b.

The two seemingly very similar ligands, Indoline-2-carboxylate and Indole-2-carboxylate (IndoliH = Indoline-2-carboxylic acid (**A**); IndolH=Indole-2-carboxylic acid (**B**)), were thus explored as *N,O*-donor systems for rhodium(I) carbonyl systems. Dicarbonyl complexes, [Rh(Indoli)(CO)$_2$] (**A1**) and [Rh(Indol)(CO)$_2$] (**B1**), were obtained directly from RhCl$_3$·xH$_2$O via the in situ synthesis of $\mu$-[RhCl(CO)$_2$]$_2$/[Rh$^{I}$Cl$_2$(CO)$_2$]$^-$ [7,19,20], as illustrated in Scheme 2. From these, triphenylphosphine complexes could be obtained (**A2**, **B2**), which were utilized as reactants for the oxidative addition of iodomethane thereto, yielding **A3** and **B3**/**B5**).

**Scheme 1.** General scheme for the proton dissociation of Indoline-2-carboxylic acid (**a**) and Indole-2-carboxylic acid (**b**) in order to enable the formation of complexes **A1**, **A2**, **B1,** and **B2**, respectively (see Scheme 2).

**Scheme 2.** General scheme for the synthesis of the compounds in this study (starting from the $\mu$-[Rh$^I$Cl(CO)$_2$]$_2$/[Rh$^I$Cl$_2$(CO)$_2$]$^-$, generated in situ from RhCl$_3\cdot$xH$_2$O) and associated ligands utilized [**A1**–**A4**: Indoline-2-carboxylic acid (IndoliH; **A**); **B1**–**B4**: Indole-2-carboxylic acid (IndolH; **B**)]. Note that (i) the dinuclear species' Rh(I) metal centers are distinguished by **1** and **2**, respectively; (ii) Indol'H$_2$ = IndolH = Indole-2-carboxylic acid (thus, Indole-2-carboxylic acid can be effectively doubly deprotonated, losing both its carboxylic and pyrone-nitrogen protons, such as in **B2**); (iii) the absolute geometry of the methyl and iodido entities following the oxidative addition of iodomethane (MeI) is not 100% defined, although current information in hand from the literature indicates that, due to steric effects, five-membered metallocycles such as those used in this study primarily yield a cis-alkyl product, which typically results from a three-centered transition state [9,11,17,21].

The compounds were obtained in good yields and could be appropriately characterized by IR, UV/Vis, and NMR spectroscopy ($^1$H-, $^{13}$C{$^1$H}-, and $^{31}$P{$^1$H}) as well as X-ray diffraction analysis, as discussed in Section 3 and reported in experimental (Section 4).

## 2.2. X-ray Crystallography

This study presents the solid state structure of an Indol rhodium(I) complex of the type [Rh$^1$(Indol')(CO)(PPh$_3$)Rh$^2$(CO)(PPh$_3$)$_2$]·(CH$_3$COCH$_3$) (**B2a**), where Indol'H$_2$: = IndolH and Indol'= Indole$^{2-}$, i.e., the dianion of Indole-2-carboxylic acid following the double deprotonation of the carboxylic acid entity, as well as the nitrogen on the pyrrole. Complex **B2a** was isolated following the addition of triphenylphosphine to [Rh(Indol')(CO)$_2$], which results in a carbonyl substitution from the rhodium coordination sphere, and which is isolated as an acetone adduct. The Vaska-type complex trans-[Rh(CO)(I)(PPh$_3$)$_2$]·(CH$_3$COCH$_3$) (**B5a**) is produced following the oxidative addition of iodomethane to [Rh$^1$(Indol')(CO)(PPh$_3$) Rh$^1$(CO)(PPh$_3$)$_2$]·(CH$_3$COCH$_3$) (**B2a**) and recrystallization from acetone. These two structures were characterized by means of single-crystal X-ray crystallography and are described herein, with basic data given in Table 1 and Figure 1.

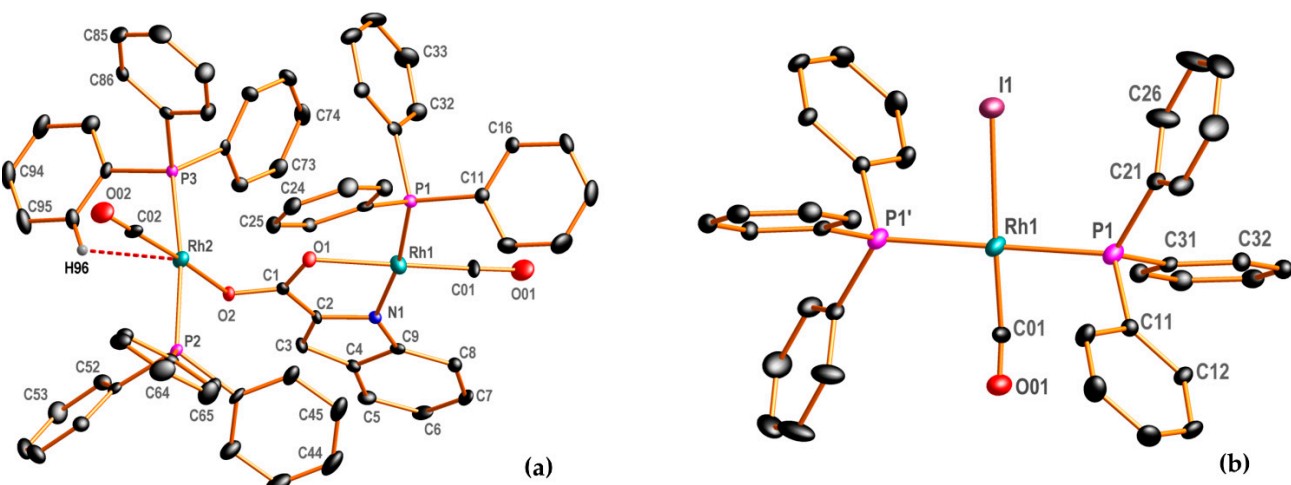

**Figure 1.** DIAMOND perspective view of (**a**) [Rh$^1$(Indol')(CO)(PPh$_3$)Rh$^2$(CO)(PPh$_3$)$_2$] (**B2**), where Indol' = Indole$^{2-}$, i.e., the dianion of Indole-2-carboxylic acid, generated following double deprotonation of the carboxylic acid entity and the pyrrole-nitrogen; (**b**) trans-[Rh(CO)(I)(PPh$_3$)$_2$] (**B5**), with a 50/50% positional disorder (not shown) occurring between the CO and Iodido ligands, with the CO oxygen situated exactly on the iodide (50% probability displacement ellipsoids). For the phenyl rings, the first digit indicates ring number, and the second digit indicates the position of the carbon atom in the ring. Apart from H96, all other hydrogen atoms, as well as the acetone solvates, have been omitted for clarity.

**Table 1.** Crystal data and refinement parameters of [Rh$^1$(Indol')(CO)(PPh$_3$)Rh$^2$(CO)(PPh$_3$)$_2$]·(CH$_3$COCH$_3$) (**B2a**) and trans-[Rh(CO)(I)(PPh$_3$)$_2$]·(CH$_3$COCH$_3$) (**B5a**).

| Identification Code | B2a [a] | B5a |
|---|---|---|
| Empirical formula | C$_{65}$H$_{50}$NO$_4$P$_3$ Rh$_2$ | C$_{43}$H$_{42}$IO$_2$P$_2$Rh |
| Formula weight | 632.93 | 898.51 |
| Temperature (K) | 100(2) | 100(2) |
| Wavelength (Å) | 0.71073 | 0.71073 |
| Crystal system, space group | Triclinic, $P\bar{1}$ | Monoclinic, $P2_1/c$ |
| Unit cell dimensions | | |
| a (Å) | 14.097(2) | 11.974(5) |
| b (Å) | 14.280(2) | 20.289(5) |
| c (Å) | 17.942(3) | 8.335(5) |

**Table 1.** *Cont.*

| Identification Code | B2a [a] | B5a |
|---|---|---|
| α (°) | 93.451(6) | 90.000(5) |
| β (°) | 112.987(5) | 98.311(5) |
| γ (°) | 115.164(6) | 90.000(5) |
| Volume (Å$^3$) | 2896.4(8) | 2003.6(2) |
| Z | 2 | 2 |
| Density$_{calc}$ (g cm$^{-3}$) | 1.451 | 1.489 |
| μ (mm$^{-1}$) | 0.705 | 1.313 |
| F(000) | 1292 | 904 |
| Crystal size (mm$^3$) | 0.196 × 0.240 × 0.414 | 0.163 × 0.189 × 0.471 |
| Theta range (°) | 2.381 to 28.00 | 3.183 to 28.00 |
| Index ranges | −14 ≤ h ≤ 14 | −15 ≤ h ≤ 15 |
| | −14 ≤ k ≤ 14 | −26 ≤ k ≤ 26 |
| | −17 ≤ l ≤ 17 | −8 ≤ l ≤ 11 |

[a] Indol$^{2-}$ = Indole-2-carboxylato dianion.

The crystal structures for **B2a** and **B5a** were determined at 100 K and the selected bond distances and angles are reported in Tables 2 and 3, respectively.

**Table 2.** Bond lengths [Å] and angles [°] for [Rh$^1$(Indol′)(CO)(PPh$_3$)Rh$^2$(CO)(PPh$_3$)$_2$]·(CH$_3$COCH$_3$) (**B2a**).

| Bond Lengths (Å) | | | | | |
|---|---|---|---|---|---|
| Rh(1)-C(01) | 1.784(5) | P(2)-C(41) | 1.826(5) | O(2)-C(1) | 1.268(6) |
| Rh(1)-N(1) | 2.065(4) | P(2)-C(61) | 1.835(5) | N(1)-C(2) | 1.384(7) |
| Rh(1)-O(1) | 2.094(3) | P(3)-C(81) | 1.827(5) | C(2)-C(1) | 1.453(7) |
| Rh(1)-P(1) | 2.2626(14) | P(1)-C(21) | 1.837(5) | C(3)-C(2) | 1.373(7) |
| Rh(2)-C(02) | 1.800(6) | P(2)-C(51) | 1.822(5) | C(4)-C(3) | 1.413(7) |
| Rh(2)-O(2) | 2.080(3) | O(01)-C(01) | 1.162(6) | C(4)-C(5) | 1.411(8) |
| Rh(2)-P(2) | 2.3355(14) | O(02)-C(02) | 1.149(6) | C(6)-C(5) | 1.367(8) |
| Rh(2)-P(3) | 2.3415(14) | O(1)-C(1) | 1.280(6) | C(7)-C(6) | 1.406(8) |
| P(1)-C(11) | 1.828(5) | C(9)-N(1) | 1.370(7) | C(8)-C(7) | 1.386(8) |
| P(1)-C(31) | 1.832(5) | C(9)-C(8) | 1.408(7) | C(9)-C(4) | 1.431(7) |
| Rh(2)-H96 | 2.786(1) | N(1)–O(1) | 2.654(1) | H(14)-H(94) | 16.09(3) |
| **Bond Angles (°)** | | | | | |
| C(01)-Rh(1)-N(1) | 98.0(2) | N(1)-Rh(1)-P(1) | 171.10(12) | C(7)-C(8)-C(9) | 117.5(5) |
| C(01)-Rh(1)-O(1) | 176.1(2) | C(11)-P(1)-Rh(1) | 116.46(17) | C(3)-C(2)-N(1) | 112.4(4) |
| N(1)-Rh(1)-O(1) | 79.32(15) | N(1)-Rh(1)-O(1) | 79.32(15) | C(3)-C(2)-C(1) | 132.0(4) |
| C(01)-Rh(1)-P(1) | 90.07(17) | C(21)-P(1)-Rh(1) | 119.01(17) | N(1)-C(2)-C(1) | 115.6(4) |
| N(1)-Rh(1)-P(1) | 171.10(12) | C(1)-O(2)-Rh(2) | 120.0(3) | C(8)-C(7)-C(6) | 122.0(5) |
| O(1)-Rh(1)-P(1) | 92.41(10) | C(1)-O(1)-Rh(1) | 114.4(3) | C(5)-C(6)-C(7) | 121.2(5) |
| C(02)-Rh(2)-O(2) | 174.9(2) | N(1)-C(9)-C(8) | 129.6(5) | C(6)-C(5)-C(4) | 118.8(5) |
| C(02)-Rh(2)-P(2) | 91.37(16) | N(1)-C(9)-C(4) | 109.8(4) | O(2)-C(1)-O(1) | 122.4(4) |
| O(2)-Rh(2)-P(2) | 88.66(10) | O(01)-C(01)-Rh(1) | 177.0(5) | O(2)-C(1)-C(2) | 120.0(4) |
| C(02)-Rh(2)-P(3) | 86.54(16) | C(9)-N(1)-Rh(1) | 141.7(3) | O(1)-C(1)-C(2) | 117.6(4) |
| O(2)-Rh(2)-P(3) | 92.71(10) | C(2)-N(1)-Rh(1) | 112.3(3) | C(2)-C(3)-C(4) | 106.0(4) |
| P(2)-Rh(2)-P(3) | 171.80(5) | O(02)-C(02)-Rh(2) | 177.3(5) | C(9)-N(1)-C(2) | 105.5(4) |
| C(02)-Rh(2)-O(2) | 174.9(2) | C(5)-C(4)-C(9) | 119.8(5) | C(8)-C(9)-C(4) | 120.6(5) |
| C(01)-Rh(1)-O(1) | 176.1(2) | C(3)-C(4)-C(9) | 106.3(4) | C(5)-C(4)-C(3) | 134.0(5) |

**Table 3.** Bond lengths [Å] and angles [°] for *trans*-[Rh(CO)(I)(PPh$_3$)$_2$]·(CH$_3$COCH$_3$) (**B5a**).

| Bond Lengths (Å) | | | |
|---|---|---|---|
| I(1)-Rh(1) | 2.7103(7) | P(1)-C(11) | 1.820(3) |
| Rh(1)-C(01) | 1.725(5) | P(1)-C(31) | 1.822(3) |
| Rh(1)-P(1) | 2.3178(9) | P(1)-C(21) | 1.828(3) |
| Rh(1)-C(01) #1 | 1.725(5) | O(01)-C(01) | 0.995(5) |
| **Bond Angles (°)** | | | |
| I(1)-C(01)-Rh(1) | 169.6(6) | C(11)-P(1)-Rh(1) | 110.67(9) |
| C(01)-Rh(1)-P(1) #1 | 94.01(18) | C(21)-P(1)-Rh(1) | 119.53(9) |
| C(01) #1-Rh(1)-P(1) #1 | 85.99(18) | C(31)-P(1)-Rh(1) | 113.53(9) |
| C(01)-Rh(1)-P(1) | 85.99(18) | P(1) #1-Rh(1)-I(1) #1 | 87.60(3) |
| C(01) #1-Rh(1)-I(1) | 176.2(2) | C(31)-P(1)-C(11) | 107.50(12) |
| P(1) #1-Rh(1)-I(1) | 92.40(3) | C(31)-P(1)-C(21) | 101.23(13) |
| P(1)-Rh(1)-I(1) | 87.59(3) | C(11)-P(1)-C(21) | 103.20(13) |

#1 Symmetry operator: (1 − x, 1 − y, 1 − z).

### 2.3. Spectroscopic Evaluation of the Oxidative Addition of Iodomethane to Rh(Indoli)(CO)(PPh$_3$)] (**A2**) and [Rh$^1$(Indol')(CO)(PPh$_3$)Rh$^2$(CO)(PPh$_3$)$_2$] (**B2**)

The solutional behavior of the complexes in the presence of iodomethane as a substrate was next evaluated by FT-IR, $^{13}$P NMR, and UV/vis spectroscopies. The monomeric complex [Rh(Indoli)(CO)(PPh$_3$)] (**A2**) was first investigated, followed by the dinuclear complex [Rh$^1$(Indol')(CO)(PPh$_3$)Rh$^2$(CO)(PPh$_3$)$_2$] (**B2**) to determine the solution characteristics and, eventually, the reactivities of the three different Rh(I) metal centers.

#### 2.3.1. Rh(Indoli)(CO)(PPh$_3$)] (**A2**)

- IR Results

Figure 2 illustrates the dynamics [22] observed within the [Rh(Indoli)(CO)(PPh$_3$)] (**A2**) complex when reacted with iodomethane. A clear decrease in the reactant is observed, coinciding with the formation of an IR band at 2054 cm$^{-1}$, which is typical of a Rh(III)-alkyl species (**A3**) [4].

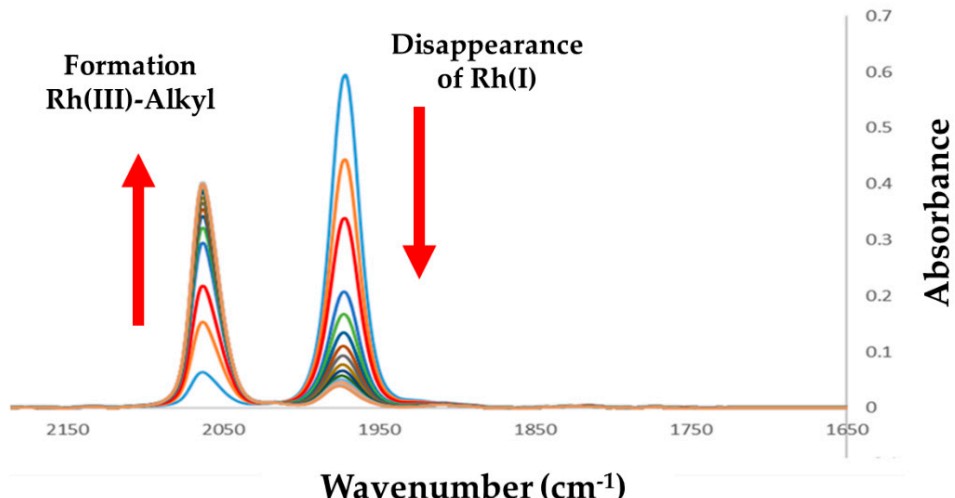

**Figure 2.** Successive IR spectra illustrating the oxidative addition of iodomethane to [Rh(Indoli)(CO)(PPh$_3$)] (**A2**) ($\nu_{(CO)}$ =1976 cm$^{-1}$) and the formation of the Rh$^{III}$-alkyl **A3** ($\nu_{(CO)}$ =2054 cm$^{-1}$); [Rh] = 0.01 M, [MeI] = 1.0 M, $\Delta$t = 120 s; Total time = 30 min.

A notable observation is the seemingly non-complete disappearance of the Rh(I) species as illustrated in Figure 2, and the absence of the formation of a Rh(III)-acyl species (with a typical $\nu_{(CO)}$ of around 1700–1720 cm$^{-1}$).

- **${}^{31}$P NMR study**

Figure 3 illustrates the corresponding dynamics observed via ${}^{31}$P NMR within [Rh(Indoli)(CO)(PPh₃)] (**A2**) when reacted with iodomethane compared to those noted in the infrared spectrum (Figure 2). Again, a clear systematic decrease in the reactant is observed ($\delta$ = 41.7 ppm; ${}^{1}J_{(Rh\text{-}P)}$ = 169 Hz), coinciding with the formation of two doublets ($\delta$=30.5 ppm, ${}^{1}J_{(Rh\text{-}P)}$ = 124 Hz, and $\delta$ = 29.5 ppm; ${}^{1}J_{(Rh\text{-}P)}$ = 124 Hz), typical of alkyl species [4], with the species represented by $\delta$ = 30.5 ppm being the dominant and the final Rh(III)-alkyl complex.

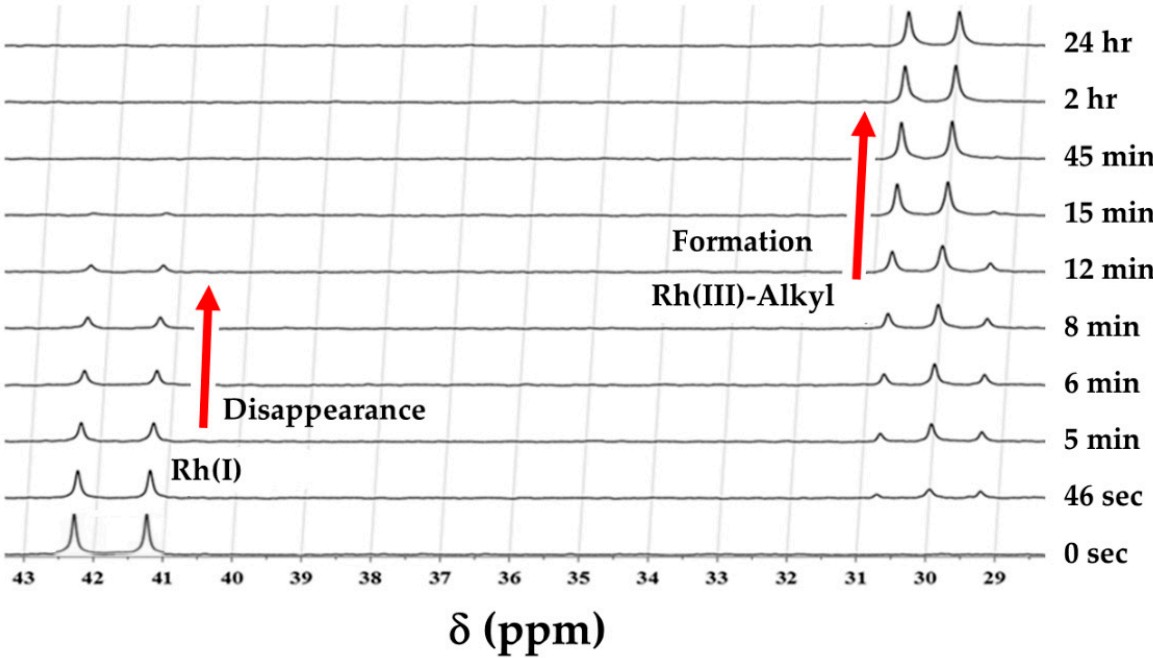

**Figure 3.** Successive ${}^{31}$P NMR spectra illustrating the oxidative addition of iodomethane to [Rh(Indoli)(CO)(PPh₃)] (**A2**) in dichloromethane at 25 °C. [Rh] = 0.02 M and [MeI] = 1.0 M. A minor Rh(III)-alkyl isomer (doublet at 29.5 ppm) is observed, which disappears after ca. 15 min, never exceeds ca 5–10% of the major isomer, and does not noticeably contribute to the reaction; thus, it has been ignored further in this study. Total time = 24 h.

### 2.3.2. [Rh¹(Indol')(CO)(PPh₃)Rh²(CO)(PPh₃)₂] (**B2**)

- **IR Study**

Contrary to that observed for the Indoline complex **A2** reported in Section 2.3.1, a completely different behavior was noted for [Rh¹(Indol')(CO)(PPh₃)Rh²(CO)(PPh₃)₂] (**B2**). The typical reaction progress is shown in Figure 4 below.

In Figure 4, a new signal of the Vaska fragment at ca. 1980 cm⁻¹ is seemingly formed. However, the deconvolution of the original spectrum before the addition of iodomethane clearly shows a shoulder in **B2**, which may be attributed to a Vaska fragment (Rh²(I)) from the start, when the complete molecule [Rh¹(Indol')(CO)(PPh₃)Rh²(CO)(PPh₃)₂] was still intact. This is additionally confirmed by the solid-state ATR IR as illustrated in Figure 4c.

- **NMR study**

Repetitive ${}^{31}$P{¹H} NMR scans (Figure 5) of the oxidative addition of iodomethane observed from IR spectroscopy in Figure 4 confirmed the behavior noted by the latter, where the rate of disappearance of Rh¹(I) ($\delta$ = 30.09 ppm; ${}^{1}J_{Rh\text{-}P}$ = 132 Hz) and the rate of appearance of Rh^{III}-alkyl ($\delta$ = 27.2 ppm; ${}^{1}J_{Rh\text{-}P}$ = 122 Hz) are equal within the experimental error. The rate of disappearance of Rh¹(I) is relatively fast and difficult to monitor reproducibly by quantitative ${}^{31}$P{¹H} NMR spectroscopy, but it was still successfully evaluated.

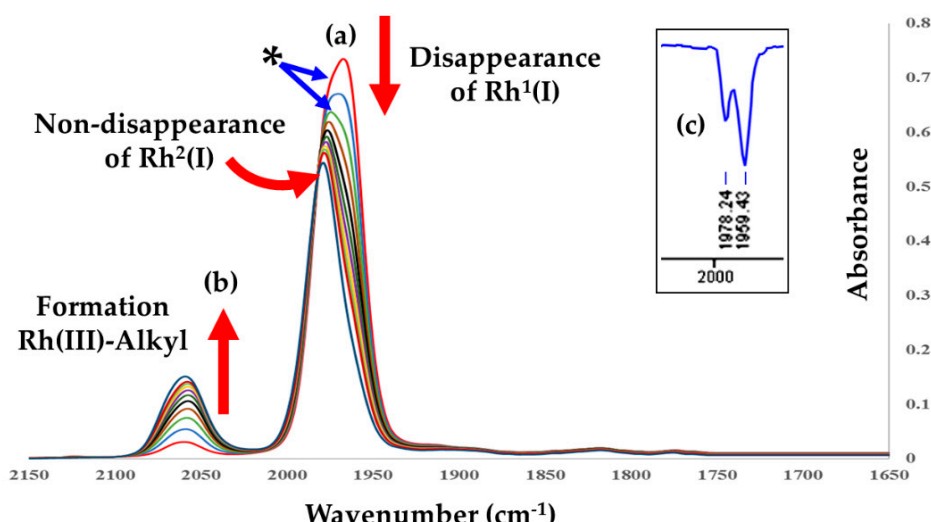

**Figure 4.** Successive infrared spectra illustrating the oxidative addition of iodomethane to [Rh$^1$(Indol')(CO)(PPh$_3$)Rh$^2$(CO)(PPh$_3$)$_2$] (**B2**), showing (**a**) the disappearance of the signal at $\nu$(CO) = 1967 cm$^{-1}$ (representing the 'sum' of the two Rh fragments, Rh$^1$(I) and Rh$^2$(I); also note the shoulder at the asterisk as indicated) and (**b**) the simultaneous definition of (i) the Vaska fragment at 1980 cm$^{-1}$ and (ii) the appearance of the Rh(III)-alkyl species at ($\nu$(CO) = 2060 cm$^{-1}$), in dichloromethane at 25 °C. [**B2**] = 0.0058 M, [MeI] = 0.064 M, and $\Delta t$ = 120 s; Total time = 20 min. Insert (**c**) illustrates the similar wavenumber range in the solid state IR ATR spectrum, and the two $\nu$(CO) peaks associated with Rh$^1$(I) and Rh$^2$(I) are more clearly defined at $\nu$ = 1959 and 1978 cm$^{-1}$, respectively (see also Figure S4 in Supplementary Materials).

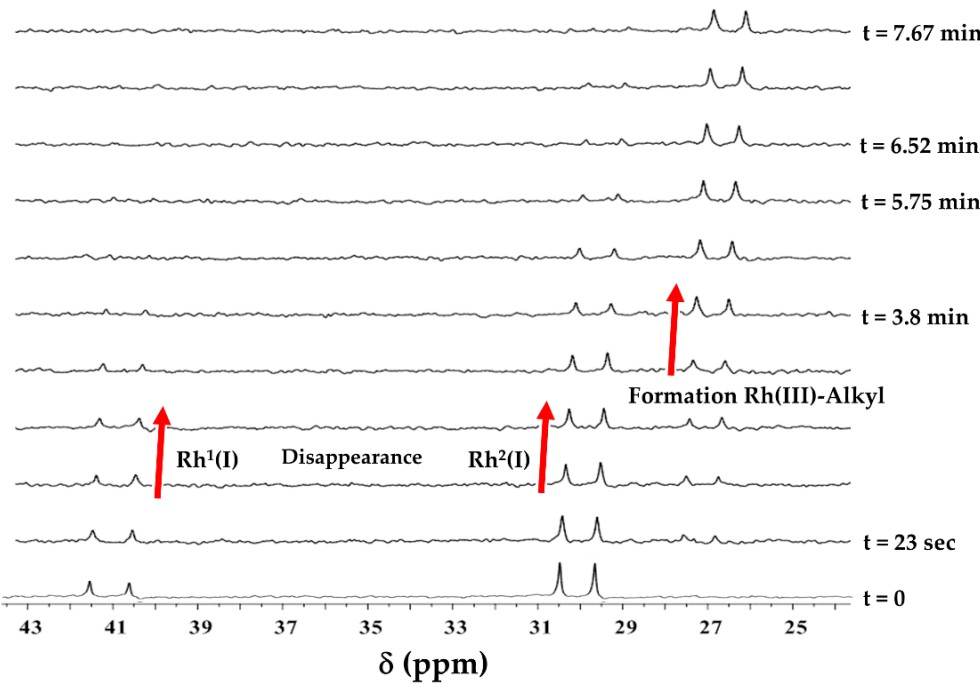

**Figure 5.** Observed $^{31}$P{$^1$H} NMR spectra illustrating the oxidative addition of iodomethane to [Rh$^1$(Indol')(CO)(PPh$_3$)Rh$^2$(CO)(PPh$_3$)$_2$] (**B2**) in dichloromethane at 25 °C. Spectra show the disappearance of the Rh$^1$(I) and Rh$^2$(I) sites and the formation of the Rh$^{III}$-alkyl (progress as per red arrows). Total peak integrals were obtained against time for 98 spectra. However, only selected spectra are plotted here to illustrate the decrease/increase in peaks (the number of scans = 12), $k_{obs}$ = (5.6 ± 0.2) × 10$^{-3}$s$^{-1}$ for Rh$^1$(I)/Rh$^2$(I), and (5.62 ± 0.17) × 10$^{-3}$s$^{-1}$ for Rh$^{III}$-alkyl. The spectra were referenced externally to 85% H$_3$PO$_4$ ($\delta$ = 0 ppm). [**B2**] = 0.0058 M; [MeI] = 0.064 M.

A summary of chemical shifts, the first-order coupling constants, and $k_{obs}$ for **B2** is given in Table 4 for this reaction.

### 2.3.3. Summary of the Oxidative Addition of Iodomethane to Rh(Indoli)(CO)(PPh$_3$)] (**2A**) and [Rh$^1$(Indol')(CO)(PPh$_3$)Rh$^2$(CO)(PPh$_3$)$_2$] (**B2**) as Monitored by Different Spectroscopies

- Rh(Indoli)(CO)(PPh$_3$)] (**A2**)

The formation of Rh(III)-alkyl was monitored by both IR and $^{31}$P NMR time-resolved spectroscopy, wherein the species' resonances were respectively analyzed. The relationship regarding the absorbance values/NMR peak integrals vs. time of the oxidative addition of iodomethane to [Rh(Indoli)(CO)(PPh$_3$)] in dichloromethane at 25 °C was now fitted to the first-order exponential given in Equation (1) (Par 4.2), and the relative reaction progress is illustrated by the four different data set fits in Figure 6.

The good fits obtained by all the data sets illustrated in Figure 6 enabled the oxidative addition of iodomethane to [Rh(Indoli)(CO)(PPh$_3$)] (**A2**) to be reliably studied (i) as a function of [MeI] and (ii) with respect to temperature. A summary of the chemical shifts, the first-order coupling constants, and the $k_{obs}$ of the oxidative addition of iodomethane to **A2** is given in Table 4.

Both the FT-IR and $^{31}$P-NMR studies agree and confirm the oxidative addition of iodomethane to **A2**, with the first reaction step being the formation of the Rh(III)-alkyl species. The $^{31}$P-NMR results show that the Rh(I) species is completely depleted at 45 min, while the Rh(III)-alkyl signal grew at the same rate.

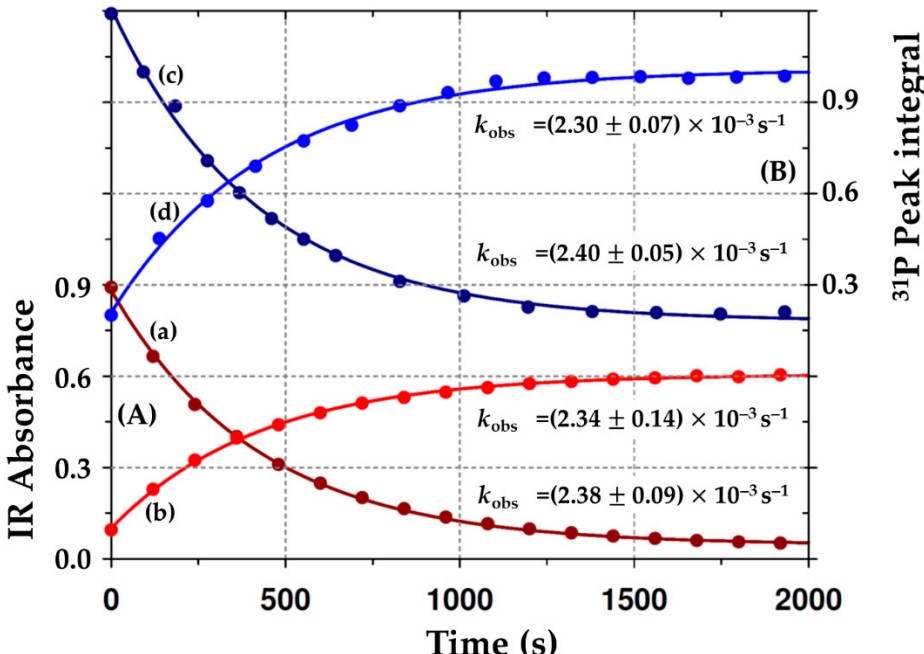

**Figure 6.** First-order fits to Equation (1) (Section 4.2) of the kinetic process at 25 °C in dichloromethane defined by the disappearance of the [Rh(Indoli)(CO)(PPh$_3$)] (**A2**) species, and the concurrent formation of the Rh(III)-alkyl species (**A3**). (A) In Red: IR Absorbance changes vs. time from Figure 2: (a) Disappearance of Rh(I); (b) formation of Rh(III)-acyl; ([**A2**] = 0.01 M; [MeI] = 1.0 M). (B) In blue: $^{31}$P{$^1$H} peak integral changes vs. time from Figure 3: (c) Disappearance of Rh(I); (d) formation of Rh(III)-acyl; ([**A2**] = 0.02 M; [MeI] = 1.0 M). The data points represent the observed IR absorbance values/$^{31}$P NMR peak integrals, while the lines give the least-squares data fits to yield the pseudo first-order rate constants.

**Table 4.** Summary of chemical shifts, coupling constants, and preliminary kinetic data in dichloromethane for the oxidative addition of iodomethane to [Rh(Indoli)(CO)(PPh$_3$)] (**A2**) and [Rh$^1$(Indol′)(CO)(PPh$_3$)Rh$^2$(CO)(PPh$_3$)$_2$] (**B2**).

| Species | IR Data | | $^{31}$P{$^1$H} NMR Data | | |
| --- | --- | --- | --- | --- | --- |
| | $\nu_{CO}$ (cm$^{-1}$) | $k_{obs}$ (s$^{-1}$) | $\delta$ (ppm) | $^1J_{(Rh\text{-}P)}$ (Hz) | $k_{obs}$ (s$^{-1}$) |
| **A2** [a,b] | | | | | |
| Rh(I) | 1967 | $(2.38 \pm 0.09) \times 10^{-3}$ [c] | 41.7 | 169 | $(2.40 \pm 0.05) \times 10^{-3}$ [c] |
| Rh$^{III}$-alkyl | 2054 | $(2.34 \pm 0.14) \times 10^{-3}$ [c] | 30.3 | 124 | $(2.30 \pm 0.07) \times 10^{-3}$ [c] |
| **B2** [d,e] | | | | | |
| Rh$^1$(I) | 1967 | $(5.56 \pm 0.12) \times 10^{-3}$ [f] | 41.0 | 150 | $(5.50 \pm 0.02) \times 10^{-3}$ [f] |
| Rh$^2$(I) | 1980 [d] | ″ | 30.0 | 133 | ″ |
| Rh$^1$(III)-Alkyl | 2060 | $(5.50 \pm 0.04) \times 10^{-3}$ [f] | 27.2 | 122 | $(5.52 \pm 0.02) \times 10^{-3}$ [f] |
| **B5** | 1968 | - | 29.0 | 126 | - |

[a] Figure 2; [b] Figure 3; [c] Figure 6; [d] Figure 4; [e] Figure 5; [f] Figure 7.

[Rh$^1$(Indol′)(CO)(PPh$_3$)Rh$^2$(CO)(PPh$_3$)$_2$] (**B2**)

Again, as for the Indoli complex **A2**, the time-resolved experiments via IR and $^{31}$P{$^1$H} NMR spectroscopy facilitated the kinetic evaluation of the Rh(III)-alkyl formation from the [Rh$^1$(Indol′)(CO)(PPh$_3$)-Rh$^2$(CO)(PPh$_3$)$_2$] (**B2**) dinuclear species, wherein the relative species' resonances were respectively analyzed. The time-resolved IR data for the oxidative addition of iodomethane to (**B2**) in dichloromethane at 25 °C, as illustrated in Figure 4, were fitted to the first-order exponential given in Equation (1) (Section 4.2), and the relative reaction progress is illustrated in Figure 7A. Thus, well-behaved kinetics were also observed for **B2** when monitored by infrared spectroscopy.

Moreover, by monitoring the $^{31}$P{$^1$H} NMR signal versus time as shown in Figure 5, well-behaved first-order kinetics predicted by Equation (1) for the formation of the Rh(III)-alkyl product were also observed as shown in Figure 7B, yielding virtually identical rate constants for the disappearance of the reactant, and the corresponding formation of the Rh(III)-alkyl product.

Finally, it should be noted that both IR and $^{31}$P NMR spectroscopy require fairly large concentrations (tens of millimolar) to allow for reproducible spectra, particularly when the reactions are fast. Thus, UV/vis spectroscopy, which allows for much lower concentrations to be used (a few millimolar, or even fractions thereof) is preferred when large numbers of kinetic runs are required, e.g., in detailed temperature and iodomethane concentration evaluations. Therefore, UV/vis spectroscopy was utilized, but always, when appropriate, in conjunction with IR and $^{31}$P NMR, as illustrated in Figure 7.

A summary of the chemical shifts, the first-order coupling constants, and $k_{obs}$ values for the [Rh$^1$(Indol′)(CO)(PPh$_3$)Rh$^2$(CO)(PPh$_3$)$_2$] (**B2**) complex is given in Table 4. A low concentration of iodomethane was necessitated due to the rapid reaction, and only a small amount of Rh$^{III}$-alkyl was formed from the two Rh(I) reactants (Figure 5). Nevertheless, even at low concentrations of MeI and Rh(I) starting materials, the data from the IR and NMR spectroscopy still correlate very well.

Interestingly, the [Rh(Indoli)(CO)(PPh$_3$)] (**A2**) complex showed no signal for a Rh(III)-acyl species as a final product from the potential migratory CO-insertion of the coordinated CO into the Rh-Me bond, even after prolonged experimental time frames (>24 h), despite being a common feature following many oxidative addition reactions of typical Rh(*L,L′*-Bid)(CO)(PR$_3$)] entities (see Scheme 2) [7,12,13]. The same observation was made for **B2**. Thus, the kinetics are significantly simplified, as illustrated in Scheme 3. Moreover, the exploratory results obtained from the IR/$^{31}$P{$^1$H} NMR experiments, as described in Section 2.3.3 for the kinetic process described in Scheme 3, could thus be used to analyze the kinetics further in more expanded detail.

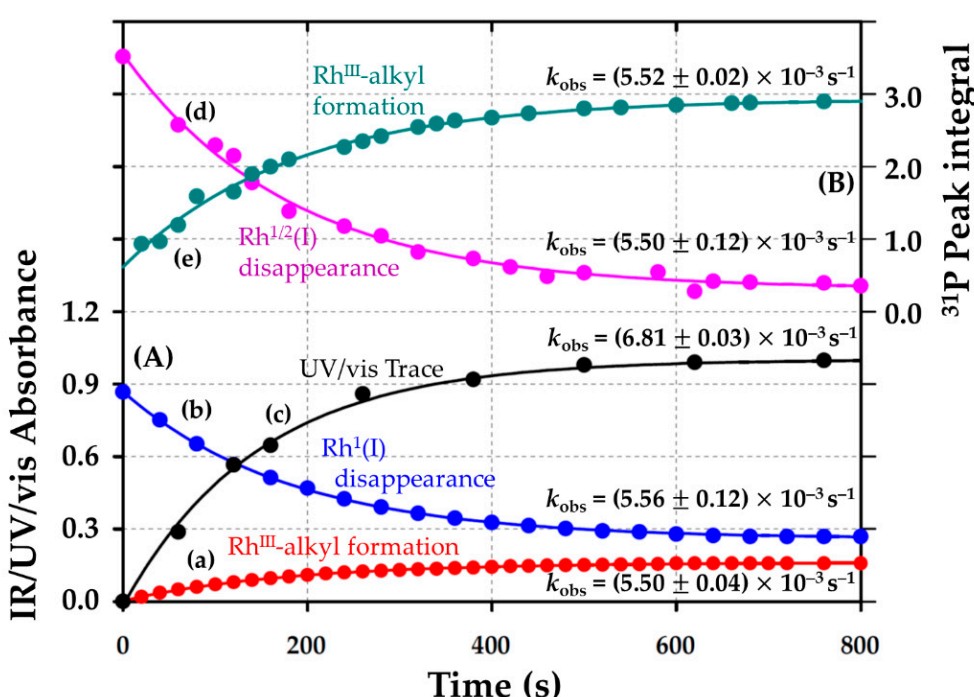

**Figure 7.** Typical first-order fits ($k_{obs}$ values given on graph) to Equation (1) of: (A) Absorbance changes as a function of time (see Figure 4): (a) IR: disappearance of **B2**; the $Rh^1(I)$ fragment; (b) IR: formation of the $Rh^{III}$-alkyl species in dichloromethane at 25 °C ([**B2**] = 0.0058 M and [MeI] = 0.064 M); (c) UV/Vis time trace in dichloromethane at 25 °C, λ = 450 nm, [**B2**] = 8.35 × 10$^{-4}$ M, and [MeI] = 0.088 M (when $k_{obs}$ normalized to [MeI] = 0.064 M, $k_{obs}$ = 5.00(3) × 10$^{-3}$ s$^{-1}$). (B) $^{31}$P{$^1$H} NMR Integrals as a function of time (see Figure 5) for the (d) disappearance of the $Rh^1(I)$ species and (e) formation of the $Rh^{III}$-alkyl species and of **B5** ([**B2**] = 0.0058 M; in dichloromethane at 25 °C; [MeI] = 0.064 M).

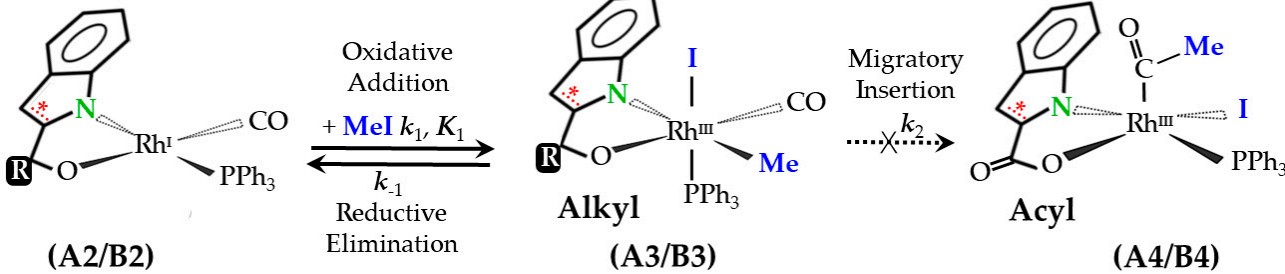

**Scheme 3.** General scheme [7] for the kinetic study of the oxidative addition of iodomethane to [Rh$^I$(*N,O*-Bid)(CO)(PPh$_3$)] complexes, showing the potential successive migratory insertion step. Note: (i) **A2–A4**: *N,O*-BidR = Ind<u>oline</u>-2-carboxylate (*without* olefinic bond (asterisked) in pyrrole ring, and R = carboxylate ketonic oxygen 'O=='); (ii) **B2–B4**: *N,O*-BidR = Ind<u>ole</u>-2-carboxylate (with olefinic bond in pyrrole ring, and with R = Vaska-type fragment—[ORh$^2$(CO)(PPh$_3$)$_2$]). A solvent pathway is not considered since it could not be formally verified in this study, although solvent effects are always present [4]. Additionally, note that although the migratory insertion step is indicated, it was not detected in the current study (shown with dotted arrow), as was observed on some occasions [23].

Notably, the formation of the Rh(III)-alkyl species in both **A2** and **B2** seems to be quite fast when compared to the literature and reactions with different bidentate systems [21,24–28].

## 3. Discussion

### 3.1. Synthesis

The synthesis of these rhodium complexes generally emanates from using the Rh(I) precursor(s) $\mu$-[Rh$^1$Cl(CO)$_2$]$_2$/[Rh$^1$Cl$_2$(CO)$_2$]$^-$, which, upon the addition of the mono-charged bidentate ligand, generate the corresponding dicarbonyl rhodium(I) complexes [4,11,19,20,29–31]. The formation of these dicarbonyl species can easily be traced by the two prominent IR stretching frequency signals at 2083 and 1987 cm$^{-1}$ in response to the two coordinated carbonyl ligands (see, e.g., Supplementary Materials Figure S1). The stoichiometric reaction of a variety of tertiary phosphine derivatives often facilitates the production of asymmetric square planar complexes, sometimes forming two isomers in solution. In the current study, triphenylphosphine was used as a monodentate ligand to substitute (only) one carbonyl ligand, yielding only one product due to the highly asymmetric Indol and Indoli bidentate ligands. The obtained light yellow precipitates were easily confirmed by one prime IR stretching frequency of one coordinated carbonyl at 1960 cm$^{-1}$ and the phosphine resonance frequency of 40.9 ppm, confirming the product to be that of the [Rh$^1$(Indoli)(CO)(PPh$_3$)] (**A2**) complex (see Scheme 2, Figures 2 and 3). The other product and reactant complexes (**A3**, **B1**–**B3**) have been successfully synthesized and confirmed as characterized by FT-IR, NMR, UV/Vis, and elemental analysis.

### 3.2. X-ray Crystallography

3.2.1. Comparison of B2a Data with the Literature

The surprising and novel dinuclear configuration, showing the Indol ligand to act *ambidentately* as a bridging entity, is very rare, only being identified once in the past for a completely different functionalized thiourea-type bidentate ligand [32].

The Indole ligand coordinates with metal ions as a bidentate ligand via the carboxylate oxygen and pyrrole nitrogen atoms to form a five-membered metallocycle, as well as a monodentate ligand coordinated with the oxygen atom [33–35]. There are a few crystal-structural reports of Indole-2-carboxylic acid ligands coordinated with different metal complexes in the literature (nine structures) [36], but not with Rh(I), and all these structures contain Indole-2-carboxylic acid coordinated as a monodentate ligand (only in this study is it coordinated as a bidentate as well as a monodentate ligand). The dianion formed by the double deprotonation of the Indole-2-carboxylic acid (Indol$^{2-}$) ligand displays an unprecedented and unexpected mode of coordination with Rh(I) by forming a bridging ligand to two rhodium centers [complex **B2**, Figure 1a].

The dinuclear complex is simplified as [Rh$^1$(Indol')(CO)(PPh$_3$)Rh$^2$(CO)(PPh$_3$)$_2$] (**B2**), where the Rh$^1$ center exhibits the expected $\kappa^2N,O$ coordination by the Indole-2-carboxylate ligand. However, although *N,O* atoms bind to the Rh$^1$ atom in the more familiar bidentate $\kappa^2N,O$ fashion, the carboxylate keto-oxygen atom coordinated ($\kappa O'$) to the Rh$^2$ atom. Hence, the Rh$^2$ center can be considered as forming a 'Rh(I) Vaska type analogue' of the common type *trans*-[Rh$^2$(PPh$_3$)$_2$(CO)(R)], where R = Cl$^-$ in the typical Ir(I) complex [37].

Selected bond distances and angles for [Rh$^1$(Indol')(CO)(PPh$_3$)Rh$^2$(CO)(PPh$_3$)$_2$]·(CH$_3$COCH$_3$) (**B2a**) are given in Tables 2 and 3 and Table S1 (Supplementary Materials). Kemp et al. [32] had determined the structure of a complex similar to **B2**, i.e., [Rh(nbnpt)(CO)(PPh$_3$)Rh(CO)(PPh$_3$)$_2$]·(CH$_3$COCH$_3$), which differs with respect to its bidentate ligand, utilizing *N*-benzoyl-*N′*-phenylthiourea, nbnptH [32]. Table S1 presents some crystallographic data of the two structures.

It is clear from Table 2 that there is a distortion from the ideal square planar geometry in complex **B2**, as well as in the complex published by Kemp et al. as evidenced by the angles around the coordination polyhedron, which differ from 90° and 180°. The P2-Rh2-P3 bond angle of 171.78(4)° for complex **B2** and the 174.78(1)° for the complex described by Kemp et al. are both smaller than 180°. The Rh2-P2 and Rh2-P3 bond distances of 2.3356(1) and 2.3419(1) Å for complex **B2**, and the 2.328(3) Å and 2.330(3) Å, respectively, for the complex by Kemp et al., are relatively long, but normal for a trans-P-Rh-P moiety [8].

The Rh1-P1 bond distance of 2.2623(1) Å in **B2** is slightly shorter than the corresponding distance of 2.275(3) Å for the complex described by Kemp et al.

Interestingly, the H14–H94 distance between the outermost H-atoms on the periphery of **B2** (and upon including the H-atom radii) is >1.7 nm and indicates that **2B** is a 'molecular nanomaterial'.

### 3.2.2. Comparison of **B5a** Data with the Literature

The structure of [Rh(CO)(I)(PPh$_3$)$_2$] (**B5**) has previously been reported in a study by Basson et al. [38] at 298 K, without the acetone solvate. Thus, **B5a** as isolated in this study is a polymorph of Basson's structure. A comparison between the structure of this study's complex **B5** and the structure by Basson et al. is given in Table S2 (Supplementary Materials).

A distortion from the ideal square planar geometry was observed in complex **B5**, as evidenced by the small I-Rh-P angle of 87.59(3)°, C01-Rh-P of 85.99(18)°, and the I-Rh-C01 angle of 176.2(2)°.

Rh-P1 and Rh-P1i have the same bond distance of 2.3178(9) Å (i refers to the symmetry-generated atom (1−x, 1−y, and 1−z) in complex **B5**), while for the complex published by Basson et al., 2.336 (2) Å and 2.316 (2) Å, respectively, were reported. These distances are relatively longer than those of the [Rh(*L*,*L*'-Bid)(CO)(PPh$_3$)] complexes [39], (where *L*,*L*'-Bid = different moncharged bidentate ligands) but normal for a trans-P-Rh-P moiety [4,40]. Nevertheless, the Rh-C bond distances of 1.725(5) Å for complex **B5**, as well as the C=O bond of 1.00(1) Å (shorter; overshadowed by the massive iodide), are in reasonable agreement with the bond distances of related compounds [7,41]. The Rh-I bond distances of 2.7103(7) Å for complex **B5** and 2.683(1) Å for Basson et al.'s complex are in good agreement with the related literature complex [42].

Basson et al. showed that the Rh complex crystallized in the monoclinic $P2_1/n$ system (Z = 4) and did so without a disorder along the I-Rh-CO axis [38]. The current study determined the space group of the structure of **B5** as monoclinic $P2_1/c$ with *a disorder* along the I-Rh-CO axis. The cell parameters were different to those reported by Basson et al. and may be attributed to the non-standard $P2_1/n$ space group and/or due to the high temperature at which the complex published by Basson et al. was collected (ambient temperature). As indicated, **B5a** is, therefore, a polymorph of that published by Basson et al. [38].

Crucially, **B5a** was obtained *directly* from the oxidative addition of iodomethane to **B2**, albeit after >24 h, which is clearly confirmed as (i) is a subsequent product of the reaction and since (ii) the Rh$^2$(I)-Vaska type fragment in **B2** undergoes substitution, but *not* oxidative addition, even after an extended period.

### *3.3. Spectroscopic Study*

#### 3.3.1. [Rh(Indoli)(CO)(PPh$_3$)] (**A2**)

As indicated in Section 2, illustrated in Figures 2 and 3, and summarized in Figure 6 and Table 4, typical first-order fits to Equation (1) for the time-resolved IR and $^{31}$P{$^1$H} NMR data were obtained and allowed for a detailed analysis of the oxidative addition to **A2**. The kinetics are straight-forward, as previously observed in a number of other analogous Rh(I) systems in the literature [4,7,11,13,23], and will be further discussed in context in Section 3.4.

#### 3.3.2. [Rh$^1$(Indol')(CO)(PPh$_3$)Rh$^2$(CO)(PPh$_3$)$_2$] (**B2**)

The IR data show that the reaction consists of a fairly rapid oxidative addition step on the [Rh$^1$(Indol')(CO)(PPh$_3$)Rh$^2$(CO)(PPh$_3$)$_2$] (**B2**) complex to form a Rh$^1$(III)-alkyl species (Figure 4; kinetic traces in Figure 7). Figure 4 adequately illustrates how the oxidative addition of iodomethane proceeds, and it is observed that the total dynamics are clearly different from those exhibited by **A2**.

Firstly, the IR vibrations corresponding to both the Rh$^1$ and the Rh$^2$ fragments' decrease ((a) $\nu_{(\text{Rh1})\text{CO}}$ = 1967 cm$^{-1}$: completely disappearing; (b) $\nu_{(\text{Rh2})\text{CO}}$ = 1975–1980 cm$^{-1}$: shifting only a few cm$^{-1}$ to a higher wavenumber).

Secondly, this coincides exactly with the appearance of a Rh(III)-alkyl fragment ($\nu_{(RhIII\text{-}alkyl)CO}$ = 2060 cm$^{-1}$). The largest effect is observed on the Rh$^1$-site, while the change in absorbance in the Rh$^2$(I)-fragment is much less pronounced.

The observations in the IR spectra (Figure 4) are exactly mimicked in the $^{31}$P{$^1$H} spectra (Figure 5), and the entire process can be summarized as follows:

- A fairly rapid oxidative addition equilibrium step causes the Rh$^1$(I) fragment to form an oxidized Rh$^{III}$-alkyl entity at the Rh$^1$(*N,O*-Indol) part, i.e., [Rh$^1$(*N,O*-Indol)(CO)(PPh$_3$)(Me)(I)] (**B3**).
- This is usually followed by the much slower carbonyl insertion still at Rh$^1$, i.e., the conversion of the corresponding Rh$^1$(III)-alkyl fragment to a corresponding [Rh$^1$(*N,O*-Idol)(COMe)-(PPh$_3$)(I)] (**B5**) Rh$^1$(III)-acyl species. However, this is not observed on the time scale utilized in the current study.
- The Rh$^2$(I)-Vaska-type fragment, on the other hand, is *cleaved off*, and does not undergo appreciable oxidative addition, as shown in the literature [43]. Additional evidence of the effect wherein the Rh$^2$(I) fragment does not undergo appreciable oxidative addition comes from the fact that the corresponding trans-[Rh(PPh$_3$)$_2$(I)(CO)] (**B5**) has been isolated from the final solution even after 1 week (see Section 4.3.7).
- There is some uncertainty as to exactly when the Rh$^2$(I)-Vaska type fragment is cleaved off, since the product Rh(III)-alkyl (**B3** in Schemes 2 and 3) with a signal at δ = 27.2 ppm (d, $^1J_{(Rh\text{-}P)}$ = 122 Hz) is close yet slightly different from the final signal of **B5** at δ 28.9 ppm (d, $^1J_{(Rh\text{-}P)}$ = 126.8 Hz), although the integral value of the Rh(III)-alkyl is close to three equivalents of PPh$_3$ (see the $^{31}$P NMR traces in Figure 6 pointing to an average, potentially dynamic equilibrium that might exist between the three different PPh$_3$ sites in the 'Rh(III)-alkyl' (**B3**) species.

### 3.4. Detailed Kinetic Study: Variation of [MeI] and Temperature

Once the dynamics within **B2** had been properly defined, as given in Section 3.3, the oxidative addition of iodomethane to [Rh(Indoli)(CO)(PPh$_3$)$_2$] (**A2**) and [Rh$^1$(Indol')(CO)(PPh$_3$)-Rh$^2$(CO)(PPh$_3$)$_2$] (**B2**) could be further reliably analyzed in more detail.

Both complexes (that is, **A2** as such, and then the Rh$^1$(I) fragment of **B2**) essentially follow the same simplified pathway as illustrated in Scheme 2 and as confirmed by FT-IR, $^{31}$P{$^1$H} NMR, and UV/Vis spectroscopy. However, as indicated, migratory insertion (Rh(III)-acyl formation) was detected for neither **A2** or **B2**.

UV/vis spectroscopy requires the use of far fewer of the reactants **A2** and **B2** and thus enables a more extended variation of the different experimental parameters; consequently, it was primarily used for the [MeI] variation and temperature study. The observed pseudo first-order rate constant data (absorbance vs. time), used to determine the effect of [MeI] and temperature on the oxidative addition reaction, as well as the second order rate constants for the oxidative addition $k_1$, were fitted to Equations (1)–(3) (Section 4.2).

### 3.4.1. [Rh(Indoli)(CO)(PPh$_3$)] (**A2**)

The oxidative addition of iodomethane to Rh(I) complexes frequently progresses as an equilibrium reaction, as presented in Scheme 3, wherein the rate of Rh(III)-alkyl species formation is dependent on the relative concentration of MeI. For example, the oxidative addition of [MeI] to the [Rh(acac)(CO)(PR$_3$)] (acacH = general β-diketones and analogs, with PR$_3$ different tertiary phosphine ligands) complexes regularly leads to linear relationship plots with non-zero intercepts [8,11–14].

In the current study, Figure 8 illustrates a similar typical linear relationship plot of the observed pseudo first-order rate constant against [MeI], corresponding to the formation of Rh(III)-alkyl species. The solid lines in Figure 8 represent the least-squares fits of the $k_{obs}$ data versus [MeI] at three different temperatures—4.5, 14.7, and 26.1 °C—and the corresponding values obtained from a global fit [23,44]. Prominent intercepts are also observed.

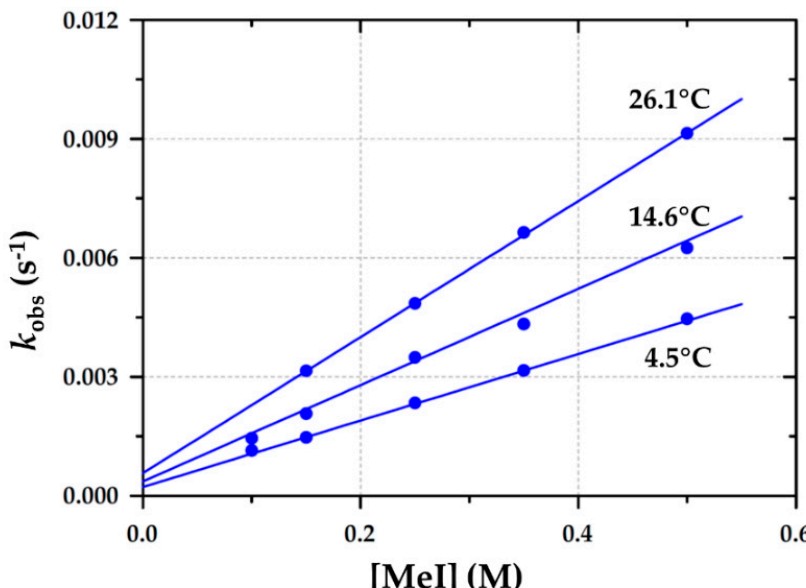

**Figure 8.** Temperature and [MeI] dependence of the pseudo first-order rate constant for the formation of the Rh$^{III}$-alkyl from [Rh(Indoli)(CO)(PPh$_3$)] (**A2**) via time-resolved UV/vis spectroscopy in dichloromethane; ($\lambda$ = 450 nm) and [**A2**] = $3.0 \times 10^{-3}$ M.

As illustrated in Figure 8, the results are consistent with the rate expression of a simple two-step reversible reaction as given in Equation (2) (Section 4.2) and Scheme 3, which is in agreement with iodomethane's oxidative addition to different Rh(I) systems in the literature [7,8,17].

The disappearance of **A2** and the formation of the Rh(III)-alkyl product **A3** was thus monitored with IR, NMR, and UV/vis spectroscopy as illustrated in Section 2.3, which gave virtually identical observed rate constants within the experimental error. The results obtained from Figure 8 as fitted to Equation (2) yielded the second-order forward rate constant, $k_1$, and are summarized in Table 5.

**Table 5.** Rate and equilibrium constants and activation parameters at different temperatures for the oxidative addition of iodomethane in dichloromethane to (a) [Rh(Indoli)(CO)(PPh$_3$)] (**A2**) and (b) [Rh$^1$(Indol')(CO)(PPh$_3$)Rh$^2$(CO)(PPh$_3$)$_2$] (**B2**).

| | Rh(Indoli)(CO)(PPh$_3$)] (A2) [a,b] | | | Activation Parameters | | | |
|---|---|---|---|---|---|---|---|
| Temp (°C) | $10^2 k_1$ (M$^{-1}$ s$^{-1}$) | $10^4 k_{-1}$ (s$^{-1}$) | $K_1$ [c] (M$^{-1}$) | $\Delta H^{\neq}$ (kJ mol$^{-1}$) | $\Delta S^{\neq}$ (J K$^{-1}$ mol$^{-1}$) | $\Delta G^{\neq}$ [d] (kJ mol$^{-1}$) | % $\Delta S^{\neq}$ to $\Delta G^{\neq}$ [d] |
| 4.5 | 0.83 ± 0.01 | 2.5 ± 0.4 | 33 ± 5 | 20 ± 2 [e] | −210 ± 6 [e] | 83 ± 3 | 76 |
| 14.7 | 1.18 ± 0.04 | 3 ± 1 | 39 ± 13 | | | | |
| 26.1 | 1.71 ± 0.01 | 5.8 ± 0.5 | 29 ± 2 | 21 ± 1 [f] | −209 ± 4 [f] | 83 ± 2 | 75 |
| | [Rh$^1$(Indol')(CO)(PPh$_3$)Rh$^2$(CO)(PPh$_3$)$_2$] (B2) [g,h] | | | | | | |
| 7.1 | 1.70 ± 0.04 | 5 ± 3 | | 66 ± 2 [e] | −40 ± 6 [e] | 78 ± 3 | 15 |
| 14.5 | 3.41 ± 0.01 | 0 | >100 [i] | | | | |
| 24.8 | 8.6 ± 0.01 | 5 ± 5 | | 73.0 ± 1.2 [f] | −21 ± 4 [f] | 79 ± 2 | 8 |
| 37.1 | 30.8 ± 0.07 | 0 | | | | | |

[a] Figure 8; [b] Figure 6; [c] Equation (2), assuming the intercept = $k_{-1}$; [d] Ref. [45]; [e] Ref. [13]; Logarithmic plots, Supplementary Materials: Figures S18 and S19; [f] Global fit of all $k_{obs}$ vs. T vs. [MeI] data [23,44]; [g] Figure 9; [h] Figure 8; [i] Estimated based on large esd's of y-intercepts.

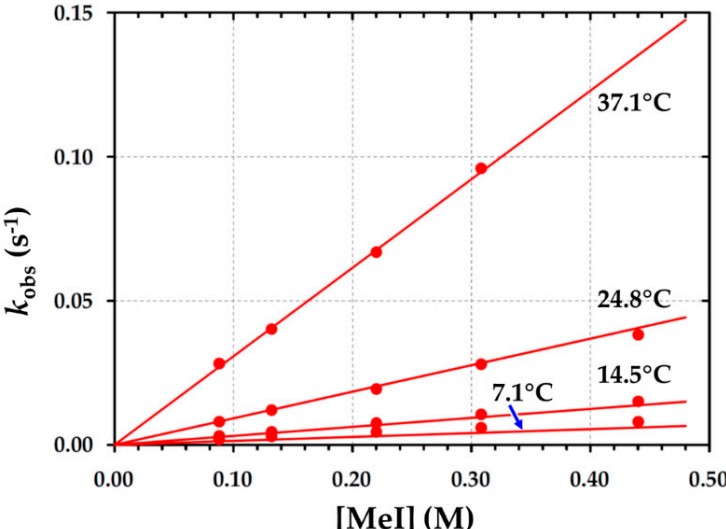

**Figure 9.** Temperature and [MeI] dependence of the pseudo first-order rate constant for the formation of $Rh^{III}$-alkyl in $[Rh^1(Indol')(CO)(PPh_3)Rh^2(CO)(PPh_3)_2]$ (**B2**) via time-resolved UV/vis spectroscopy in dichloromethane; ($\lambda$ = 450 nm) and [**B2**] = $8.35 \times 10^{-4}$ M.

If the oxidative addition to **A2** proceeds to completion, it, in principle, excludes the possibility that the intercept could represent a significant reverse reaction, i.e., reductive elimination ($k_{-1}$). This implies large positive values for the equilibrium constant as denoted by $K_1$ if the Rh(III)-alkyl species is quite stable, indicating that the non-zero intercept is indicative of a *solvent* pathway. This is supported by Figure 3, which indicates the *complete disappearance* of the Rh(I) reactant (no reverse reaction).

However, the opposite is suggested by Figure 3, where there is clearly some unreacted Rh(I) at the end of the reaction, suggesting that not all the Rh(I) reactant has been consumed. This seemingly contradictory evidence left us assuming the presence of a potential solvent pathway but did not provide sufficient proof of whether the intercept indicates reductive elimination or a solvent pathway.

The experimental temperature variation study of the oxidative addition of iodomethane to **A2** (as illustrated in Figure 8) enabled the determination of the activation parameters (Table 5) from which an Eyring plot [11] was constructed (Supplementary Materials: Figure S18) or via a global fit of all the data points in Figure 8 [23,44].

### 3.4.2. $[Rh^1(Indol')(CO)(PPh_3)Rh^2(CO)(PPh_3)_2]$ (**B2**)

UV/Vis spectroscopy was further used to analyze the oxidative addition of iodomethane to **B2** and produced virtually identical results to those found by IR and NMR spectroscopy in dichloromethane at 25 °C (Figure 7 and Table 4). The temperature dependence of the reaction is illustrated in Figure 9, while the rate constants obtained are reported Table 5.

This complex follows the same mechanism for oxidative addition as described in Scheme 3, and forms only one major isomer, **B3**, negating any formation of isomerization Rh(III)-acyl products during the oxidative addition of iodomethane. Still, the formation of **B5** took place as indicated, but did not prevent the detailed study of the oxidative addition of iodomethane at the $Rh^1$(I) site in **B2**.

The small intercept value (near zero) of $k_{-1}$ suggests that neither reductive elimination nor a solvent pathway are operative in the overarching oxidative addition of iodomethane to **B2**. Therefore, this implies a large value for the equilibrium ($K_1 = k_1/k_{-1}$), which significantly favors a forward reaction. As a result of some spectral cross-interference in the Rh(I) and $Rh^{III}$-alkyl signals, a combined effect was observed at a high concentration of iodomethane, which, in turn, led to low absorption readings at advanced reaction times. The equilibrium constant ($K_1$) was not determined precisely because of the uncertainty in

most of the values of $k_{-1}$. A reasonable estimate, however, is that $K_1 > 100$ M$^{-1}$, which is typical of less steric complexes and more readily accessible by the entering MeI.

The experimental temperature variation study of the oxidative addition of iodomethane to **B2,** as illustrated in Figure 9, again enabled the determination of the activation parameters from which an Eyring plot could be constructed [11] (Supplementary Material: Figure S19) or via a global fit of the data in Figure 9 [23,44]. As given in Table 5, the current iodomethane oxidative addition study yielded a significantly smaller negative activation entropy [$\Delta H^{\neq}$ = (73.0 $\pm$ 1.2) kJ mol$^{-1}$ and $\Delta S^{\neq}$ = (−21 $\pm$ 4) J K$^{-1}$mol$^{-1}$], wherein the $\Delta S^{\neq}$ contribution to the free energy of activation, $\Delta G^{\neq}$, at 25 °C is only ca. 10%. This might be due to the bulkiness of **B2,** which results in a more complex transition state and may be interpreted as indicating a more interchange-associated type of mechanism.

### 3.4.3. Kinetic Correlation of [Rh$^1$(Indoli)(CO)(PPh$_3$)] (**A2**) with [Rh$^1$(Indol')(CO)(PPh$_3$)-Rh$^2$(CO)(PPh$_3$)$_2$] (**B2**)

The oxidative addition of iodomethane to both **A2** and **B2** shows a direct conversion of the rhodium(I) starting material to the rhodium(III) alkyl species as the final product, without suggesting a Rh(III)-acyl species as a final product. Thus, a migratory CO-insertion of the coordinated methyl into the Rh-CO bond, even after prolonged experimental time frames > 24 h, is not favored, as indicated by the IR and $^{31}$P NMR data.

Table 5 illustrates the activation parameters obtained from the Eyring equation as well as a global fit of all the $k_{obs}$ vs. [MeI] vs. Temperature data to the exponential form of the Eyring equation. It presents the forward second-order rate constant $k_1$ for both **A2** and **B2** at different temperatures, as well as the free energy of activation, $\Delta G^{\neq}$, at 25 °C.

It is evident that a slower reaction is observed for **A2** when compared to **B2**, wherein a larger contribution of the activation entropy [$\Delta H^{\neq}$ = (21 $\pm$ 1) kJ mol$^{-1}$ and $\Delta S^{\neq}$ = (−209 $\pm$ 4) J K$^{-1}$mol$^{-1}$] to the free energy of activation, $\Delta G^{\neq}$, is obtained (the contribution of $\Delta S^{\neq}$ to $\Delta G^{\neq}$ of ca. 75%) [45]. The significant negative value of $\Delta S^{\neq}$ for **A2** is indicative of an associative mechanism for the oxidative addition step and, subsequently, an effective entropy-controlled reaction, as found in many [Rh($L,L'$-Bid)(CO)(PPh$_3$)] complexes [4,7–9,23].

Although the oxidative addition of iodomethane to [Rh$^1$(Indol')(CO)(PPh$_3$)-Rh$^2$(CO)(PPh$_3$)$_2$] (**B2**) is expected to be influenced by the increased complexity within the molecule and potentially the added large steric effect augmented by the two trans triphenylphosphine ligands coordinated to Rh$^2$, the rate of the reaction for **B2** is actually ca. five times faster at 25 °C than **A2**. Thus, the net electronic effect of the three groups of triphenylphosphine surrounding the two rhodium metals seems to overshadow the steric effect. It also yielded a significantly smaller negative activation entropy [$\Delta H^{\neq}$ = (73.0 $\pm$ 1.2) kJ mol$^{-1}$ and $\Delta S^{\neq}$ = (−21 $\pm$ 4) J K$^{-1}$mol$^{-1}$], wherein the contribution of $\Delta S^{\neq}$ to the free energy of activation, $\Delta G^{\neq}$, at 25 °C was only ca. 10%. This is contrary to the entropy-driven oxidative addition of iodomethane to **A2** ($\Delta S^{\neq}$ contribution to $\Delta G^{\neq}$ of ca. 75%) (see Table 5). The literature assumes typical normal associative processes for the oxidative addition of iodomethane to these types of complexes and displays large negative values for $\Delta S^{\neq}$ [4,7,10,11].

These seemingly different results regarding the processes applied to the Indoline-2-carboxylate and Indole-2-carboxylate bidentate ligands explored in this study, although structurally very similar, are very interesting, notable, and illustrate the fact that small changes in ligand systems (main group elements-based) may induce significant effects in the behavior of transition metals—in this case, rhodium(I).

## 4. Materials and Methods

### 4.1. General Procedure and Instrumentation

All chemical reagents used for the syntheses were of analytical grade and obtained from Sigma-Aldrich, South Africa. Solvents were of analytical grade and were used without additional purification, and all syntheses were performed under aerobic conditions. $^1$H, $^{13}$C{$^1$H}, and $^{31}$P{$^1$H} FT-NMR solution state spectra for the complexes were recorded at

25 °C on a Bruker Avance II 600 MHz spectrometer ($^1$H = 600.28 MHz, $^{13}$C = 150.95 MHz, and $^{31}$P = 242.99 MHz), Bruker Avance III 400 MHz spectrometer ($^1$H = 400.13 MHz, $^{13}$C = 100.61 MHz, and $^{31}$P = 161.97 MHz), and Bruker Fourier NMR 300 MHz spectrometer ($^1$H = 300.18 MHz, and $^{13}$C = 75.48 MHz). Chemical shifts are reported in ppm and coupling constants in Hz. For all complexes, the deuterated solvents used for analysis are mentioned with the spectral results. A Bruker Digilab FTS 2000 Fourier-transform spectrometer (ATR) was used to record all infrared spectra of complexes in the range 600–3000 cm$^{-1}$.

A Bruker ApexII 4K CCD diffractometer was used for crystal data collection via Mo *Kα* (0.71073 Å) and *ω*-scans at 100(2) K. All reflections were merged and integrated with SAINT-PLUS [46], and SADABS [46] was used to correct Lorentz, polarization, and absorption effects. The heavy atom method was used to determine the structures and was refined through full-matrix least squares cycles using SHELX-97 [47] as part of the WinGX [48] package, with $\Sigma(\|Fo| - |Fc\|)^2$ and Olex2 [49] being minimized. All non-H atoms were refined with anisotropic displacement parameters, while H atoms were constrained to parent atom sites using a riding model (aromatic C–H = 0.95 Å; aliphatic C–H = 0.98 Å). The graphics were created using DIAMOND Visual Crystal Structure Information System software [50] with 50% probability displacement ellipsoids for non-hydrogen atoms. Elemental analyses for CHN were performed by Atlantic Microlab Inc., 6180 Atlantic Blvd., Suite M, Norcross, GA 30071 (USA).

All reagents used for the syntheses and characterization were of analytical grade and were purchased from Sigma-Aldrich, South Africa. Some reagents were used as received without further purification. All organic solvents were purified and dried according to accepted procedures.

*4.2. Kinetic Measurements*

All experiments were performed under an air atmosphere, and only pure organic solvents were used for the kinetic measurements. All $^{31}$P NMR spectra were collected on a Bruker Avance III 400 MHz NMR spectrometer in CD$_2$Cl$_2$ at 161.97 MHz. For the collection of data, a pre-selected number of scans were applied to ensure quantitative spectra, (typically 12–84 scans) using a 90° pulse for $^{31}$P (25 μs), decoupling 90° pulse for $^1$H (8 μs), 64 k data points, delay time of 2 s, and broadband H-decoupling. $^{31}$P chemical shifts are reported relative to 85% H$_3$PO$_4$ (0 ppm); positive shifts are downfield. The kinetic data reported were calibrated using CD$_2$Cl$_2$ as standard. The FT-IR spectra were recorded as liquid samples in dry organic solvents (DCM) in a NaCl cell on a Bruker Tensor 27 spectrometer in the range of 3000–600 cm$^{-1}$, equipped with a temperature cell regulator accurate within 0.3 °C. UV/Vis absorbance spectra were collected in 1.000 ± 0.001 cm tandem quartz cuvettes on a Varian Cary 50 Conc spectrophotometer, equipped with a temperature cell regulator, accurate within 0.1 °C.

The kinetic data were analyzed with the Scientist software package [51] by fitting the UV/vis-absorbance/NMR/IR data to the first-order exponential given in Equation (1), as reported previously [24,25,52] based on the simple reaction shown in Scheme 3.

The $k_{\text{obs}}$ values were calculated from plots of absorbance against time graphs using the following first-order equation with parameters A$_{\text{obs}}$, A$_0$, A$_\infty$, and t as the overall observed absorbance, initial absorbance at the start of the reaction, final absorbance, and time, respectively.

$$\text{A}_{\text{obs}} = \text{A}_\infty - (\text{A}_\infty - \text{A}_0)e^{(-\text{kobs. t})} \tag{1}$$

The oxidative addition steps were studied using different iodomethane concentrations at different temperatures. The observed pseudo first-order rate constant data were plotted against [MeI] in order to determine the rate constant for oxidative addition $k_1$ and reductive elimination or solvent pathway as intercept $k_{-1}$, fitted to Equation (2) [24,25,52].

$$k_{obs} = k_1[\text{MeI}] + k_{-1} \tag{2}$$

Under equilibrium conditions and in absence of an appreciable solvent pathway, the equilibrium constant for the reaction in Scheme 3 is given by Equation (3).

$$K_1 = k_1/k_{-1} \tag{3}$$

The Indoline and Indole Rh(I) systems [**A2** and **B2**] were studied at different temperatures to determine the effect of temperature on the iodomethane oxidative addition reaction using the Eyring equation (Equation (4); logarithmic form [53–57] and Equation (5); exponential form (for global fit) [23,44,58]).

$$\ln(k_1/T) = \ln(k/h) - \Delta H^{\neq}/RT + \Delta S^{\neq}/R \tag{4}$$

$$k_1 = (k_B T/h) \cdot e^{(\Delta S^{\neq}/R)} \cdot e^{(-\Delta H^{\neq}/RT)} \tag{5}$$

*4.3. Synthesis*

4.3.1. Synthesis of Dicarbonyl(Indoline-2-Carboxylato-κO,N)rhodium(I), [Rh(Indoli)(CO)$_2$] (**A1**)

RhCl$_3$·xH$_2$O (0.100 g; 0.380 mmol) was dissolved in 5 cm$^3$ of DMF and refluxed for 30 min. The solution was cooled to room temperature. [Rh(Indoli)(CO)$_2$] was synthesized by addition of Indoline-2-carboxylic acid (**A**; 0.093 g and 0.570 mmol) to the cooled solution. An equivalent amount (0.0470 g; 0.570 mmol) of sodium acetate was added for deprotonation. A light-yellow precipitate was formed after the addition of ice water. The precipitate was filtered off and dried (Yield: 0.070 g; 59.0%). **IR**: $\nu_{CO}$ 2083, and 1987 cm$^{-1}$. **$^1$H NMR** (400 MHz, dimethyl sulfoxide-d$_6$): 8.88 (d, J = 53.4 Hz, 1H), 7.37 – 7.23 (m, 2H), 7.17 (d, J = 19.7 Hz, 1H), 4.24 (s, 1H), and 3.33 (s, 2H). **$^{13}$C{$^1$H} NMR** (101 MHz, dichloromethane-d$_6$): 188.9 (m), 181.9 (m), 180.6 (s), 148.2 (w), 146.1 (s), 133.1 (d, $^1J$ = 30 Hz),128.3 (s), 126.2 (d, $^1J$ = 30 Hz), 118.4 (d, $^1J$ = 30 Hz), 62.1 (s), and 35.0 (m).

4.3.2. Synthesis of carbonyl(indoline-2-caboxylato-κO,N)(triphenylphosphine-κP)-rhodium(I), [Rh(Indoli)(CO)(PPh$_3$)] (**A2**)

[Rh(Indoli)(CO)(PPh$_3$)$_2$] was synthesized by dissolving [Rh(Indoline)(CO)$_2$] (0.062 g, 0.193 mmol) in 5 cm$^3$ of acetone. Triphenylphosphine (PPh$_3$) (0.050 g and 0.193 mmol) was added to the solution with stirring applied, resulting in the immediate evolution of CO gas. After a few minutes, the reaction was completed, as determined by IR spectroscopy. Ice water was added dropwise to the solution. A yellow precipitate was filtered off and dried. (Yield: 0.077 g; 61.0%). **IR**: $\nu_{CO}$ 1960 cm$^{-1}$. **$^1$H NMR** (300 MHz, dichloromethane-d$_2$): 7.80 – 7.37 (m, 15H), 7.22 (d, $^1J$ = 26.2 Hz, 2H), 6.33 (t, $^1J$ = 1H), 4.46 (t, 1H), 3.50 (d, $^1J$ = 23.6 Hz, 2H), and 2.19 (t, $^1J$ = 21.2 Hz, 1H). **$^{13}$C{$^1$H} NMR** (101 MHz; acetonitrile-d$_6$): 190.2 (m), 179.2 (s), 145.0 (w), 140.5 (s), 134.3 (s), 132.6 (s), 131.9 (s), 130.3 (s), 128.5 (s) 124.5 (s), 120.0 (s), 116.3 (d, $^1J$ = 50 Hz), 103.3 (s), 101.3 (s), 61.1 (m), and 32.5 (m). **$^{31}$P{$^1$H} NMR** (243 MHz, dichloromethane-d$_2$): 40.94 (d, $^1J$ = 167.6 Hz).

4.3.3. In Situ Synthesis/Characterization of carbonyl(indoline-2-caboxylato-κO,N)(iodido)(methyl)(triphenylphosphine-κP)-rhodium(III)] [Rh(Indoli)(I)(Me)(CO)(PPh$_3$)] (**A3**)

Typical in situ characterization of the Rh(III)-alkyl species was primarily performed by $^{31}$P NMR and Infrared spectroscopy in dichloromethane as follows: [MeI] = 0.10 M was added to a solution of **A2** (0.010 M) in dichloromethane at ca. 25 °C. Successive infrared spectra illustrating the oxidative addition of iodomethane to [Rh$^1$(Indoli)(CO)(PPh$_3$)] (**A2**) indicated the disappearance of the signal at ν(CO) = 1976 cm$^{-1}$ and yielded the Rh(III)-alkyl, [Rh(Indoli)(I)(Me)(CO)(PPh$_3$)] (**A3**) (ν(CO) = 2054 cm$^{-1}$; Figure 2).

A similar solution when monitored by $^{31}$P{$^1$H} NMR spectroscopy showed the synchronized disappearance of **A2** (δ[Rh(I)] = 41.9 ppm; Figure 3) and appearance of [Rh(Idoli)(I)(Me)-(CO)(PPh$_3$)] (**A3**) (δ[Rh(III)-alkyl] = 27.4 ppm).

### 4.3.4. Synthesis of dicarbonyl(Indole-2-carboxylato-$\kappa$O,N)rhodium(I), [Rh(Indol)(CO)$_2$] (**B1**)

RhCl$_3\cdot$xH$_2$O (0.0500 g; 0.1898 mmol) was dissolved in 5 cm$^3$ of DMF and refluxed for 30 min. The solution was cooled to room temperature. Indol-2-carboxylic acid (**B**; 0.037 g and 0.228 mmol) was added to the cooled solution. An equivalent amount (0.0187 g; 0.2277 mmol) of sodium acetate was added for deprotonation. A pink precipitate was formed after the addition of ice water. The precipitate was filtered off and dried. (Yield: 0.0415 g; 68.60%). **IR**: $\nu_{CO}$ 2075 and 1995 cm$^{-1}$. $^1$**H NMR** (300 MHz; acetone-d$_6$): 7.99 (s, 1H), 7.53 (d, $^1J$ = 7.5 Hz, 1H), 7.44 (d, $^1J$ = 8.1 Hz, 1H), 7.13 (t, 1H), 6.90 (t, 1H). $^{13}$C{$^1$H} **NMR** (75 MHz, acetone-d$_6$): 191.1 (w), 190.1 (s), 176.0 (s), 138.2(s), 131.4 (m), 129.2 (m), 123.5 (s), 119.2(s), 116.1(s), 108.6 (s), and 105.6 (m). Anal. Calcd for C$_{11}$H$_{11}$NO$_7$Rh (%): C, 35.50; H, 2.98; N, 3.76. Found: C, 35.15; H, 3.09; N, 6.03.

### 4.3.5. Synthesis of [Rh$^1$(Indol')(CO)(PPh$_3$)Rh$^2$(CO)(PPh$_3$)$_2$] (**B2**)

The dinuclear species [Rh(Indol')(CO)(PPh$_3$)Rh(CO)(PPh$_3$)$_2$] (**B2**) was synthesized by dissolving [Rh(Indol)(CO)$_2$] (0.0412g; 0.129 mmol) in 5 cm$^3$ of acetone. Triphenylphosphine (PPh$_3$) (0.0339 g; 0.1292 mmol) was added to the solution with stirring, resulting in the immediate evolution of CO gas. After a few minutes, the reaction was completed as determined by IR spectroscopy. Ice water was added dropwise to the solution. A yellow precipitate was filtered off and dried. (Yield: 0.097 g; 61.8%) Yellow cuboid crystals suitable for X-ray diffraction were obtained from recrystallization in acetone and a few drops of water.

**IR**: $\nu_{CO}$ 1978, 1959 cm$^{-1}$. $^1$**H NMR** (600 MHz, dichloromethane-d$_2$): 7.74–7.68 (m, 1H), 7.52 (dd, $^1J$ = $^1J$ = 12.2, 5.8 Hz, 2H), 7.45 (d, $^1J$ = 6.0 Hz, 2H), 7.40–7.37 (m, 1H), 7.35 (t, $^1J$ = 7.4 Hz, 1H), 7.25 (t, $^1J$ = 7.5 Hz, 2H), 6.98 (t, $^1J$ = 7.4 Hz, 1H), 6.80 (t, $^1J$ = 7.3 Hz, 1H), 6.16 (s, 1H).$^{13}$C{$^1$H} **NMR** (75 MHz, CDCl$_2$): 190.4 (m), 189.6 (m), 177.9 (s), 144.3 (s), 140.3(s), 133.7 (s), 133.2 (s), 131.2 (s), 130.4 (s), 129.1 (s), 127.1 (s), 120.1 (s), 115.5 (d $^1J$ = 76 Hz), and 102.6 (s). $^{31}$**P{$^1$H} NMR** (243 MHz, dichloromethane-d$_2$): 41.08 (d, $^1J$ = 150.2 Hz), 30.09 (d, $^1J$ = 132.9 Hz). Anal. Calcd for C$_{68}$H$_{56}$NO$_5$P$_3$Rh$_2$ (%): C, 64.52; H, 4.46; N, 1.11. Found: C, 63.85; H, 4.36; N, 1.13.

### 4.3.6. In Situ Synthesis/Characterization of carbonyl(Indole-2-carboxylato-$\kappa$O,N)(iodido) (methyl)(triphenylphosphine-$\kappa$P)rhodium(I), [Rh(Indol)(I)(Me)(CO)(PPh$_3$)] (**B3**)

Typical in situ characterization of the Rh(III)-akyl species was primarily performed by $^{31}$P NMR and Infrared spectroscopy in dichloromethane as follows: [MeI] = 0.064 M was added to a solution of **B2** (0.010 M) in dichloromethane at ca. 25 °C. Successive infrared spectra illustrating the oxidative addition of iodomethane to [Rh$^1$(Indol')(CO)(PPh$_3$)-Rh$^2$(CO)(PPh$_3$)$_2$] indicated the disappearance of the signal at $\nu$(CO) = 1967 cm$^{-1}$ (representing the 'sum' of the two Rh fragments, Rh$^1$(I) and Rh$^2$(I)) and yielded the Rh(III)-alkyl, [Rh(Indol)(I)(Me)(CO)(PPh$_3$)] (**B3**) ($\nu$(CO) = 2060 cm$^{-1}$; Figure 4). However, the simultaneous appearance of the Vaska fragment at 1980 cm$^{-1}$ was also prominent.

A similar solution when monitored by $^{31}$P{$^1$H} NMR spectroscopy showed the synchronized disappearance of **B2** (Rh fragments: $\delta$[Rh$^1$(I)] = 40.1 ppm, and $\delta$[Rh$^2$(I)] = 30.1 ppm; Figure 5) and appearance of [Rh(Indol)(I)(Me)(CO)(PPh$_3$)] (**B3**), ($\delta$[Rh(III)-alkyl] ca. 28 ppm.

### 4.3.7. Synthesis of the Vaska-Type Complex trans-[Rh(CO)(I)(PPh$_3$)$_2$] (**B5**)

The Vaska-type complex trans-[Rh(CO)(I)(PPh$_3$)$_2$] (**B5**) was obtained from the kinetic reaction of iodomethane's oxidative addition to [Rh(Indol')(CO)(PPh$_3$)Rh(CO)(PPh$_3$)$_2$] (**B2**). It was synthesized by dissolving **B2** (0.0121 g; 0.0100 mmol) in 1 cm$^3$ of acetone. Iodomethane (0.0227 g; 0.1600 mmol) was added to the aforementioned solution and the solution was kept for one week. (Yield: 0.0032 g; 40.8%.) Yellow cuboid crystals, suitable for X-ray diffraction, were obtained.

Trans-[Rh(CO)(I)(PPh$_3$)$_2$] (**B5**) was also independently synthesized by dissolving [Rh(CO)(Cl)(PPh$_3$)$_2$] (0.0050 g; 0.0072 mmol) in 1 cm$^3$ of acetone. Tetra-n-butylammonium

iodide $(CH_3CH_2CH_2CH_2)_4N(I)$ (0.0027g; 0.0072 mmol) was added to the solution with stirring applied. Yellow powder was obtained after evaporating the acetone. (Yield: 0.0039 g; 70.00%.) IR: $\nu_{CO}$ 1968 cm$^{-1}$. $^{31}P\{^1H\}$ NMR (162 MHz, dichloromethane-d$_2$) δ 28.9.0 (d, *J* = 126.8 Hz).

## 5. Conclusions

An important contribution to the coordination chemistry of rhodium(I) Indole/Indoline complexes has been presented herein. It underlines the delicate interplay between appropriately selected main group bidentate ligand systems such as Indoline-2-carboxylic acid (**B**, IndoliH) and Indole-2-carboxylic acid (IndolH, **B**) to platinum group metals, which is further manifested by a detailed study of the oxidative addition of iodomethane to rhodium(I) carbonyl triphenylphosphine complexes.

IndoliH formed [Rh(Indoli)(CO)(PPh$_3$)] (**A2**), while IndolH yielded the novel dinuclear [Rh$^1$(Indol')(CO)(PPh$_3$)Rh$^2$(CO)(PPh$_3$)$_2$] (**B2**) complex (Indol' = Indol$^{2-}$) as characterized by SCXRD. In **B2**, the Rh$^1$(I) fragment [Rh$^1$(Indol')(CO)(PPh$_3$)] (bidentate *N,O*-Indol) exhibits a distorted square-planar geometry, while Rh$^2$(I) shows a 'Vaska'-type distorted square-planar trans-[O-Rh$^2$(PPh$_3$)$_2$(CO)] configuration (bridging the carboxylate 'oxo' O atom of Indol$^{2-}$).

A detailed kinetic study of the oxidative addition of iodomethane to **A2** and **B2** via time-resolved FT-IR, NMR, and UV/Vis spectroscopy indicated only Rh(III)-*alkyl* species (**A3**/**B3**) as products; thus, no migratory insertion occurred. A variable temperature kinetic study confirmed an associative mechanism for **A2** via the general equilibrium-based pathway ($\Delta H^{\neq}$ = (21 ± 1) kJ mol$^{-1}$; $\Delta S^{\neq}$ = (−209 ± 4) J K$^{-1}$mol$^{-1}$), with a smaller contribution from a reverse reductive elimination/solvent pathway. The dinuclear complex **B2** showed MeI oxidative addition *only* at Rh$^1$(I), forming a Rh(III)-alkyl, but cleaved the bridged Rh$^2$(I) site, yielding trans-[Rh$^1$(PPh$_3$)$_2$(I)(CO)] (**5B**) as a secondary product (characterized by SCXRD).

The significantly smaller negative activation entropy [$\Delta H^{\neq}$ = (73.0 ± 1.2) kJ mol$^{-1}$; $\Delta S^{\neq}$ = (−21 ± 4) J K$^{-1}$mol$^{-1}$] suggests a more complex and potential *interchange* mechanism for the dinuclear complex **B2** (the contribution of $\Delta S^{\neq}$ to the Gibbs free energy of activation, $\Delta G^{\neq}$, is only ca. 10%), contrary to the entropy-driven and purely *associative* oxidative addition of MeI to **A2** ($\Delta S^{\neq}$ contribution to $\Delta G^{\neq}$ ± 75%).

Finally, this study allowed for important comparisons to be drawn with similar structures from the literature and illustrated the significant chemical and geometric differences between the complexes **A2** and **B2**. Consequently, these findings enabled further understanding of the physical and chemical behaviors of potential larger, dinuclear Rh(I) model catalysts and their structure/reactivity relationships and may prompt further investigation into their future applications as, e.g., applied catalysts.

**Supplementary Materials:** The complete list of supporting information can be downloaded at: https://www.mdpi.com/article/10.3390/inorganics10120251/s1, Figure S1: ATR IR spectrum of [Rh(indoli)(CO)$_2$] (A1); Figure S2: ATR IR spectrum of [Rh(indoli)(CO)(PPh$_3$)] (A2); Figure S3: ATR IR spectrum of [Rh$^1$(Indol')(CO)(PPh$_3$)Rh$^2$(CO)(PPh$_3$)$_2$] (B2); Figure S5: $^1$H NMR spectrum (DCM) of [Rh(indoli)(CO)$_2$] (A1); Figure S6: $^{13}$C NMR spectrum of the IndoliH ligand in Acetonitrile d3; Figure S7: $^{13}$C NMR spectrum (DCM) of [Rh(indoli)(CO)$_2$] (A1); Figure S8: $^1$H NMR spectrum (DCM) of [Rh(indoli)(CO)(PPh3)] (A2); Figure S9: $^{13}$C NMR spectrum (Acetone-d6) of [Rh(indoli)(CO)(PPh$_3$)] (A2); Figure S10: $^{31}$P NMR spectrum (DCM) of [Rh(indoli)(CO)(PPh$_3$)] (A2); Figure S11: $^1$H NMR spectrum (DCM) of [Rh(indol)(CO)$_2$] (B1); Figure S12: $^1$H NMR spectrum (DCM) of [Rh$^1$(Indol')(CO)(PPh$_3$)Rh$^2$(CO)(PPh$_3$)$_2$] (B2); Figure S13: $^{13}$C NMR spectrum (Acetone-d6) of IndolH; Figure S14: $^{13}$C NMR spectrum (Acetone-d6) of [Rh(indol)(CO)$^2$] (B1); Figure S15: $^{13}$C NMR spectrum (CDCl$_3$) of [Rh$^1$(Indol')(CO)(PPh$_3$)Rh$^2$(CO)(PPh$_3$)$_2$] (B2); Figure S16: $^{31}$P NMR spectrum (DCM) of [Rh$^1$(Indol')(CO)(PPh$_3$)Rh$^2$(CO)(PPh$_3$)$_2$] (B2); Figure S17: $^{31}$P NMR spectrum (DCM) of (a) trans-[RhI(CO)(PPh$_3$)$_2$] (B4) and (b) *trans*-[RhClCO)(PPh$_3$)$_2$]; Figure S18: Eyring plot: $k_1$ rate constant (DCM): iodomethane oxidative addition to [Rh(indoli)(CO)(PPh$_3$)] (A2); Figure S19: Eyring plot: OA $k_1$ rate constant to [Rh$^1$(Indol')(CO)(PPh$_3$)Rh$^2$(CO)(PPh$_3$)$_2$]

(B2) in dichloromethane.; Table S1: Comparison of selected crystallographic data in $[Rh^1(Indol')(CO)(PPh_3)Rh^2(CO)(PPh_3)_2]\cdot\cdot(CH_3COCH_3)$ (B2a) with an isostructural thiourea molecule reported by Kemp et al. [Ref see ms]; Table S2: Comparison of geometric parameters and crystal data of *trans*-$[Rh(CO)(I)(PPh_3)_2]\cdot\cdot(CH_3COCH_3)$ (B5a) and similar structure found in the literature.

**Author Contributions:** Conceptualization, A.R. and J.A.V.; methodology, A.R. and J.A.V.; software, A.R. and O.T.A.; validation, M.A.E.E., A.R. and J.A.V.; formal analysis, M.A.E.E.; investigation, M.A.E.E.; resources, G.J.S.V., A.R. and J.A.V.; data curation, M.A.E.E.; writing—original draft preparation, M.A.E.E. and O.T.A.; writing—review and editing, M.A.E.E., A.R., J.A.V. and O.T.A.; visualization, M.A.E.E., A.R. and O.T.A.; supervision, J.A.V. and A.R.; project administration, G.J.S.V., J.A.V. and A.R.; funding acquisition, G.J.S.V., J.A.V. and A.R. All authors have read and agreed to the published version of the manuscript.

**Funding:** We acknowledge the University of the Free State, Department of Chemistry, the South African National Research Foundation (SA NRF) (UID 99782) for its financial support. This includes funding under the Swiss–South Africa joint research program (SSAJRP) from the SA NRF (AR: UID: 107802), the Competitive Program for Rated Researchers of the SA NRF (AR: UID 111698), and the Competitive Program for Unrated Researchers of the SA NRF (J.A.V.: 116302).

**Data Availability Statement:** The crystallographic data sets for compounds **B2a** and **B5a** are available as Supplementary Material and from the Cambridge Structural Data Center with the following codes: CCDC 2214777 (**B2a**) 2214778 (**B5a**). Other data and spectroscopy are available as Supplementary Information [Figures S1–S16].

**Acknowledgments:** Sincere thanks are also due to Linette Twigge for the NMR data analysis. Additional acknowledgement further goes to the crystallographic data collection team at the University of the Free State in Bloemfontein, South Africa.

**Conflicts of Interest:** The authors declare no conflict of interest.

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
