# Peer review of "Structural Study of Model Rhodium(I) Carbonylation Catalysts Activated by Indole-2-/Indoline-2-Carboxylate Bidentate Ligands and Kinetics of Iodomethane Oxidative Addition"

_inorganics, doi:10.3390/inorganics10120251_

Round 1

Reviewer 1 Report

The manuscript deals with the formation of new rhodium(I)-indol- or indoline-2-carboxylate complexes and the kinetics of iodomethane oxidative addition. The chemistry is well-done, particularly interesting is the formation of dimer B2, so it merits publication in Inorganics after taking into consideration the following issues:

- The drawings in the Scheme 1 need corrections. I suggest to indicate that sodium acetate is necessary to deprotonate Indol or Indoline in the reaction with [RhCl(Co)2]2. A double bond between the nitrogen atom and the C2 of the indol must be drawn in A(1-4) and  B1, B3 and B4. For B2 the double bond must be drawn between the carbon attached to both oxigen atoms and C2 of the indol fragment.  Addition of sodium Acetate  must be introduced Moreover, notation for B4 y B5 are exchanged in relation with the discussion text.

- Is the empirical Formula of B4a correct in table1?  It seems H30 not H32. Check it please.

-In the Scheme 2, it is missed the same insaturation that has been comented for Scheme 1. Brackets with 0 indicating not charge can be deleted. The double bond drawed between C3-C4 correspond to indol-complexes but the authors refer here also to indoline-type complexes (A), so a dotted line there would be better. 

-In part 3.1,lines 332-333 it is said Rh(III) dinuclar dimer which is actually Rh(I).

-Did the authors mean elemental analysis when chemical analysis is writing? (line 344,772).

-Table 5 and 6 are not very relevant to this work and can be better suited in Supporting Information.

-I wonder If a 1H or 13C NMR espectrum can be recorded, even in situ, at least for one oxidative addition product (A3-B3) in order to confirm the presence of a rhodium-methyl ligand.

-Elemental analysis or HRMS must be provided for A1 and A2 or its absence must be justified.

-Please, recheck the 13C data of B1 and B2. In the first case carbonyl and aromatic signal are missed and the values from 29 to 28 ppm has no sense since B1 has no aliphatic carbons. In the second case carbonyl signal are lost. To which carbon are ascribed the resonance at 30.76-30.47 ppm?

In conclusion a thorough recheck of drawings and NMR data a necessary before publication.

Reviewer 2 Report

The manuscript “Structural Study of Model Rhodium(I) Carbonylation Catalysts Activated by Indole-2-/ Indoline-2-Carboxylate Bidentate Ligands and Kinetics of Iodomethane Oxidative Addition” discussed the synthesis of rhodium complexes with bidentate indole or indoline ligands and its behavior in the oxidative addition with methyl iodide. Also kinetics of formation and further transformation of product were investigated using NMR and UV time experiments.

After reading of manuscript I want to take to Authors several for points for reflection.

i)                    First of all I want to write that 30 of 66 references are articles contain A. Roodt as co-author. In my opinion it is too much…

ii)                  The manuscript is not clear to read and understand at most because of the chapters “Results” and “Discussion” are separated. I think the manuscript needs to completely rewrite and merge the chapters to “Results and Discussion”.

iii)                The next point deals with the synthetic procedure. The refluxion of ruthenium trichloride hydrate in DMF leads to anionic species [RhCl2(CO)2]- (doi:10.1016/s1387-1609(00)88534-5) but not to neutral dimeric μ-[RhCl(CO)2]2 as described authors. Also I found another article about interaction between RhCl3 x H2O and DMF (DOI:10.1134/S1070328407030074) but there did not described formation of μ-[RhCl(CO)2]2 too. The authors of reviewed manuscript in line 98 referred to own  previous article (DOI: 10.1002/ejic.201800293). But there they referred to well-known technique of Inorganic Synthesis (Doi: 10.1002/9780470132593.ch20). Thus the authors thesis that they started from “the in-situ synthesed μ-[RhCl(CO)2]2” is not absolutely correct.

iv)                Further, the manuscript has absolutely huge “Abstract” and “Conclusions” sections and its need to be drastically reduced, especially last one.

v)                  The scheme 1 is not acceptable too. It is is very information overloaded. I think the scheme 1 need to separate to several parts. Also in the text and authors described crystal structures of products B2a and B4a. Both of them are absent in the scheme 1. It looks like authors isolated two side product and described they. I think all these discussion (with tables of distances and angles) need to move to Supporting Information section. Additionally no any characteristics or details of isolation were present in Experimental section.

vi)                By the way, not any Supplementary Information attached to the manuscript!! However in line 204 authors referred to figure S4 from it's.

vii)              The compounds B4 in the text (for example in scheme 1) and in Experimental part are different.

Thus I believe that the manuscript in its current form is not suitable for publication and requires a radical revision and rewriting.

Reviewer 3 Report

Thanks for the opportunity to review the manuscript titled, “Structural Study of Model Rhodium(I) Carbonylation Catalysts Activated by Indole-2-/ Indoline-2-Carboxylate Bidentate Ligands and Kinetics of Iodomethane Oxidative Addition” by Elmakki and co-workers. The current manuscript presented the different reaction models of [Rh(Indol)(CO)2] and [Rh(Indoli)(CO)2], and the following related oxidative addition reactions of iodomethane were well investigated.

 The following comments need to be addressed.

1. Abstract needs to be revised. The experimental details usually should not be included in the Abstract section, and “All the compounds were characterized by FT-IR and NMR spectroscopy. The iodomethane oxidative addition to both A2 and B2 was studied by time-resolved FT-IR, NMR and UV/Vis spectroscopy” could be removed.

The current Abstract looks like a summary of experimental investigations, instead of an attractive and brief introduction. Please also revise the Conclusion section accordingly.

2. Introduction: A figure or scheme to show the special reactivity of rhodium(I) complexes controlled by the different ligand coordination modes.

Besides reference 5, the studies on the platinum complex from Blacquiere’s report (Dalton Trans., 2022,51, 3977-3991) may be also needed to be considered.

3. Discussions: 3.2.1 Comparison of B2a data with literature. Please provide a scheme for the structure of Kemp’s report.

Discussions: 3.2.2 Comparison of B4a data with literature. Please provide a scheme for the structure of Basson’s report.

4. Supplementary Materials are missing. 

Round 2

Reviewer 1 Report

After several changes, I find now the paper suitable for publication

Author Response

We thank the reviewer for his/her detailed comments

Reviewer 2 Report

I carefully read the response of the Authors and their comments. Regarding the mention of ruthenium instead of rhodium in one of the comments, I apologize, it is a typo.
As for everything else, I remain of my opinion that the submitted manuscript should not be published by Inorganics. The shorting of the "Abstract" and "Conclusions" by 7 and 15% looks like a mockery of common sense.
I leave the final decision to the Editor.

Author Response

We acknowledge the comment from the reviewer and will respond accordingly to the points made by the Academic Editor